# The selective estrogen receptor downregulator GDC-0810 is efficacious in diverse models of ER+ breast cancer

James D Joseph[1], Beatrice Darimont[1], Wei Zhou[2], Alfonso Arrazate[2], Amy Young[2], Ellen Ingalla[2], Kimberly Walter[3], Robert A Blake[4], Jim Nonomiya[4], Zhengyu Guan[2], Lorna Kategaya[5], Steven P Govek[6], Andiliy G Lai[6], Mehmet Kahraman[6], Dan Brigham[1], John Sensintaffar[1], Nhin Lu[1], Gang Shao[1], Jing Qian[1], Kate Grillot[1], Michael Moon[1], Rene Prudente[1], Eric Bischoff[1], Kyoung-Jin Lee[7], Celine Bonnefous[6], Karensa L Douglas[6], Jackaline D Julien[6], Johnny Y Nagasawa[6], Anna Aparicio[7], Josh Kaufman[7], Benjamin Haley[8], Jennifer M Giltnane[9], Ingrid E Wertz[5], Mark R Lackner[3], Michelle A Nannini[2], Deepak Sampath[2], Luis Schwarz[10], Henry Charles Manning[11], Mohammed Noor Tantawy[11], Carlos L Arteaga[10], Richard A Heyman[1], Peter J Rix[7], Lori Friedman[2], Nicholas D Smith[6], Ciara Metcalfe[2]*, Jeffrey H Hager[1]*

[1]Department of Biology, Seragon Pharmaceuticals, San Diego, United States; [2]Department of Translational Oncology, Genentech, South San Francisco, United States; [3]Department of Oncology Biomarker Development, Genentech, South San Francisco, United States; [4]Department of Biochemical and Cellular Pharmacology, Genentech, South San Francisco, United States; [5]Departments of Discovery Oncology and Early Discovery Biochemistry, Genentech, South San Francisco, United States; [6]Department of Chemistry, Seragon Pharmaceuticals, San Diego, United States; [7]Department of Drug Safety and Disposition, Seragon Pharmaceuticals, San Diego, United States; [8]Department of Molecular Biology, Genentech, South San Francisco, United States; [9]Department of Pathology, Genentech, South San Francisco, United States; [10]Department of Medicine and Breast Cancer Program, Vanderbilt-Ingram Cancer Center, Nashville, United States; [11]Vanderbilt University Institute of Imaging Science, Vanderbilt University, Nashville, United States

*For correspondence:
metcalfe.ciara@gene.com (CM);
jeff.hager@icloud.com (JHH)

**Abstract** ER-targeted therapeutics provide valuable treatment options for patients with ER+ breast cancer, however, current relapse and mortality rates emphasize the need for improved therapeutic strategies. The recent discovery of prevalent ESR1 mutations in relapsed tumors underscores a sustained reliance of advanced tumors on ER$\alpha$ signaling, and provides a strong rationale for continued targeting of ER$\alpha$. Here we describe GDC-0810, a novel, non-steroidal, orally bioavailable selective ER downregulator (SERD), which was identified by prospectively optimizing ER$\alpha$ degradation, antagonism and pharmacokinetic properties. GDC-0810 induces a distinct ER$\alpha$ conformation, relative to that induced by currently approved therapeutics, suggesting a unique mechanism of action. GDC-0810 has robust in vitro and in vivo activity against a variety of human breast cancer cell lines and patient derived xenografts, including a tamoxifen-resistant model and those that harbor ER$\alpha$ mutations. GDC-0810 is currently being evaluated in Phase II clinical studies in women with ER+ breast cancer.
DOI: https://doi.org/10.7554/eLife.15828.001

## Introduction

Breast cancer is the most commonly diagnosed cancer and the second leading cause of death in women (*Edwards et al., 2014*). About 75% of breast cancers express estrogen receptor alpha (ERα), a hormone-regulated transcription factor (*Dunnwald et al., 2007*). ERα-positive breast cancers typically respond well to therapy that attenuates ERα signaling, either by blocking the production of estrogens via aromatase inhibitors, or antagonizing the activity of estrogens through competitive binding of ER antagonists such as tamoxifen (*Puhalla et al., 2012*). While estrogen disrupting therapies are often effective both in the adjuvant and metastatic setting, patients frequently relapse after prolonged therapy (*Chia et al., 2008*; *Dowsett et al., 2010*; *Davies et al., 2011*; *Mauri et al., 2006*).

Though resistance to endocrine therapies frequently emerges, relapsed tumors remain dependent on ER, which is highlighted by patient responses to second and third line endocrine therapies after failure of an earlier line (*Mouridsen et al., 2009*; *Howell et al., 2005*; *Lewis and Jordan, 2005*; *Perey et al., 2007*). ER signaling re-activation in the relapsed setting can occur due to changes in ligand sensitivity and specificity, for example, resistance to tamoxifen has been proposed to be a result of the selection of tumor cells that recognize the ER:tamoxifen complex as agonistic (*Gottardis and Jordan, 1988*; *McDonnell and Wardell, 2010*; *Takimoto et al., 1999*). More recently however, a series of mutations in the ligand binding domain of ERα have been found at a high prevalence (25–40%) in relapsed, metastatic patients, though are very rare in untreated populations (*Jeselsohn et al., 2014*; *Li et al., 2013*; *Merenbakh-Lamin et al., 2013*; *Robinson et al., 2013*; *Toy et al., 2013*). These mutations have been shown in overexpression experiments to confer estrogen-independent activity to the receptor, and to reduce the potency of, for example, tamoxifen. The continued dependence of breast cancer tumors on ERα provides a strong rationale to continue to target ER in both first line and relapsed/advanced settings.

Selective ER Downregulators (SERDs) are competitive ERα antagonists that also induce a conformational shift of the receptor that results in ubiquitination and subsequent degradation of ERα, via the ubiquitin-proteasome system (*Berry et al., 2008*; *Wijayaratne and McDonnell, 2001*; *Wijayaratne et al., 1999*). The unique dual-function of SERDs (ER antagonism and depletion) may enable them to block ER signaling in cellular settings where other endocrine agents, such as tamoxifen or aromatase inhibitors have failed. Indeed, the clinical impact of fulvestrant as a treatment of recurrent, endocrine resistant disease, supports this notion (*Perey et al., 2007*).Though fulvestrant has served as an important proof of concept for the SERD approach [see also (*Ellis et al., 2015*)], it is limited by its poor pharmaceutical properties, which necessitates administration by intramuscular injection and limits the applied dose, exposure, and receptor engagement (*Robertson et al., 2001*; *van Kruchten et al., 2015*). The fulvestrant 500 mg regimen (500 mg on day 1, 14, 28; monthly thereafter) exhibited improvement in progression free survival and overall survival over the initially approved and marketed 250 mg regimen (*Di Leo et al., 2010*). The 500 mg regimen achieves higher plasma concentrations and a more rapid ascent to steady-state drug levels, which in turn is thought to result in superior modulation of ER signaling (*Kuter et al., 2012*; *Ohno et al., 2010*; *Pritchard et al., 2010*). However, the 500 mg dose does not fully saturate ER binding in patients, as inhibition of [$^{18}$F]fluoroestradiol uptake was incomplete in 38% (6/16) of patients analyzed. Importantly, this lack of receptor occupancy was associated with lack of clinical benefit (*van Kruchten et al., 2015*). Together, these data demonstrate that SERDs have the potential to provide effective and well-tolerated therapy for postmenopausal women with advanced breast cancer, and highlight the need for the development of SERDs with optimized bioavailability and pharmacokinetic properties. Here we describe a novel, potent non-steroidal ER antagonist and degrader, GDC-0810, that is orally bioavailable and has strong anti-tumor activity in endocrine-sensitive and -resistant models of ER+ breast cancer.

## Results

### Identification of GDC-0810, a novel SERD

With the goal of creating a next generation, orally bioavailable SERD, we prospectively optimized ERα degradation, antagonist activity, as well pharmacokinetic properties, starting with a triphenylalkene ER ligand scaffold (*Lai et al., 2015*). These efforts resulted in the identification of an indazole series of SERDs, including GDC-0810 (*Figure 1A*). Compounds were assessed for their ability to modulate ERα protein levels in MCF7 breast cancer cells using a quantitative In-Cell Western (ICW) immunofluorescence assay, which enabled determination of compound potency and maximal activity in a high throughput format. GDC-0810 was prioritized for further characterization based on its ability to robustly reduce cellular ERα levels within 4 hr, to levels approaching that observed with fulvestrant, with sub-nanomolar potency (*Figure 1B* and *Table 1*). The active metabolite of tamoxifen, 4-hydroxytamoxifen (4OH-tamoxifen), exhibited some activity in this assay, though it failed to reduce ERα signal to the level achieved by fulvestrant or GDC-0810. The ability of GDC-0810 to reduce steady-state levels of ERα was confirmed using a Western Blot assay (*Figure 1C*). We further demonstrated that GDC-0810-mediated ERα depletion is dependent on the 26S proteasome, since addition of the proteasome inhibitor MG132, fully blocked GDC-0810 depletion of ERα, similar to the effect of MG132 treatment in preventing fulvestrant-mediated ER turnover (*Figure 1C,D*).

In general, data generated in the ICW assay correlated well with Western Blot assays (*Figure 1—figure supplement 1*). For certain ligands which showed modest activity in the ICW assay, but did not induce ERα degradation by Western Blot, such as 4OH-tamoxifen, the observed reduction of ERα signal in the ICW assay may be the result of changes in protein conformation, protein complex formation, or subcellular localization that limit the detection of the antibody epitope in the in situ ICW assay.

In cell-free radio-ligand competitive binding assays, GDC-0810 binds both ERα and ERβ with low nanomolar affinity [*Table 1*, see also (*Lai et al., 2015*)]. In MCF7 cells GDC-0810 effectively antagonizes E2-mediated transcriptional activation of an ER reporter construct and inhibits cell proliferation with nanomolar potency and efficacy similar to that of fulvestrant and 4OH-tamoxifen (*Table 1*). GDC-0810 is selective against the other nuclear hormone receptor (NHR) family members as monitored by competitive binding and reporter activation assays (*Supplementary file 1A–C*).

### GDC-0810 induces a distinct ERα conformation versus tamoxifen and other ER therapeutics, and does not exhibit tamoxifen-like ER agonism in MCF7 cells

Nuclear receptors impart biological action through ligand-induced conformational change resulting in exposure of co-regulator interaction surfaces. To assess the ERα conformation induced by GDC-0810 relative to other SERMs and SERDs, we utilized a mammalian 2-hybrid based peptide interaction assay. This assay, run in the presence of saturating concentrations of ER ligands, differentiates ligand-induced ER conformations based on the ability of the ligand-bound receptor to interact with discriminatory peptide probes (*Iannone et al., 2004*; *Norris et al., 1999*; *Paige et al., 1999*). We included in our analysis fulvestrant, 4OH-tamoxifen, a number of additional clinically approved ER ligands, such as bazedoxifene, and also GW7604. GW7604 is an active metabolite of GW5638, an ER ligand that entered Phase I clinical trials for breast cancer, but is not currently in further clinical development. Like GDC-0810, GW5638 is a non-steroidal dual function ER antagonist and degrader (*Lai et al., 2015*; *Willson et al., 1994*, *Willson et al., 1997*). The GDC-0810:ER complex recruits a distinct set of peptides relative to the other ER:ligand complexes profiled here, with the exception of GW7604 (*Lai et al., 2015*). GDC-0810 thus induces an ER conformational profile that is different from that induced by 4OH-tamoxifen, fulvestrant and other marketed SERMs and similar to GW7604 (*Figure 2A*). Since the biological activity of ERα is determined by the conformation of the receptor and its ability to interact with other regulatory proteins, this result implies differences in the response of ERα to GDC-0810 and the clinically approved ERα therapeutics.

Tamoxifen is a Selective ER Modulator (SERM), and as such can either antagonize, or agonize ER signaling, in a gene- and cell-specific manner. Though 4OH-tamoxifen primarily antagonizes estrogen-dependent ER signaling in breast cancer cells, its partial agonistic activity has been revealed through gene expression profiling in the absence of estrogen, in MCF7 cells (*Wardell et al., 2013*).

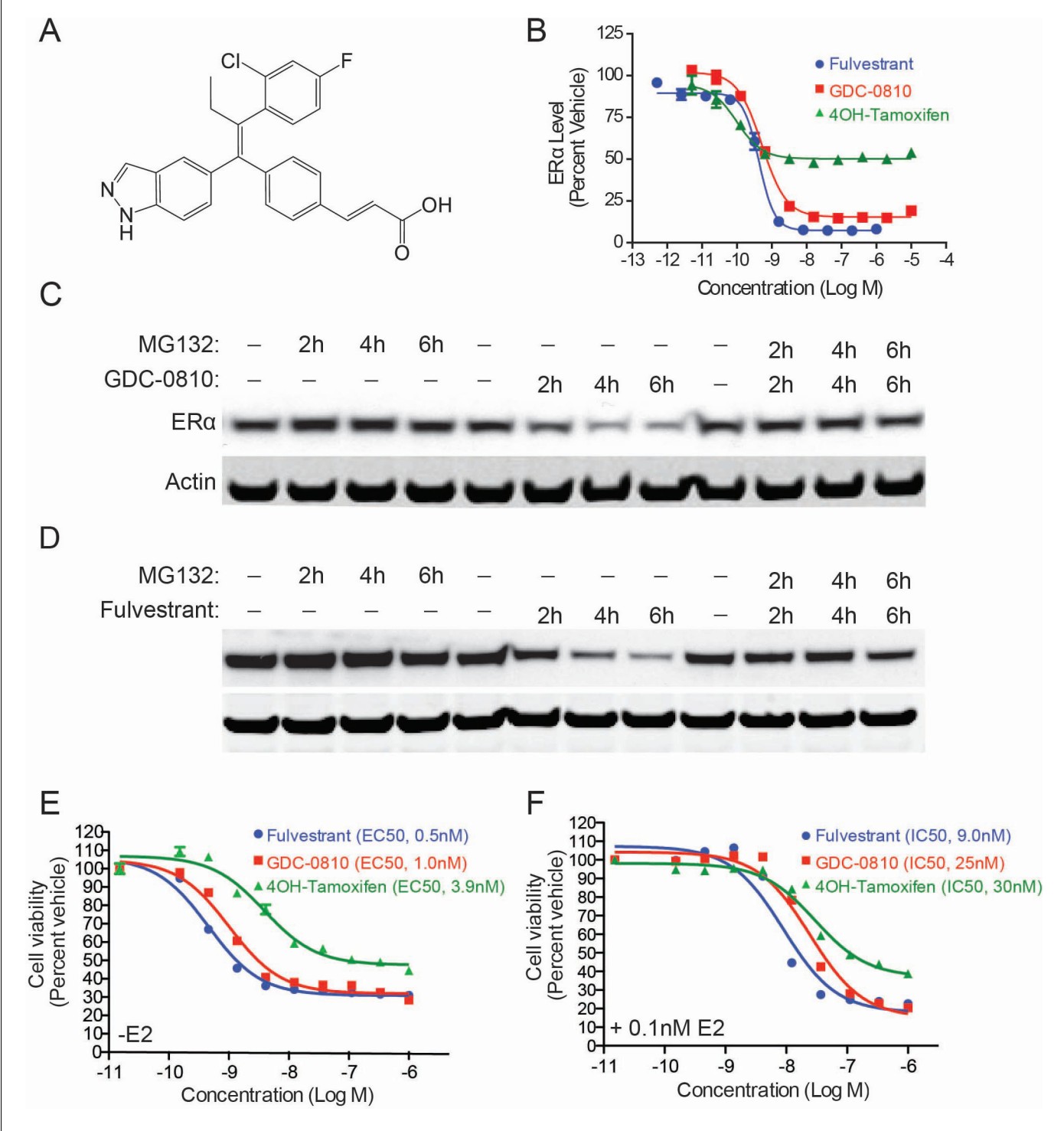

**Figure 1.** GDC-0810 induces proteasome-dependent degradation of ERα and suppresses proliferation of MCF7 cells. (**A**) GDC-0810 structure. (**B**) MCF7 ERα In-Cell Western assay comparing GDC-0810 potency to fulvestrant and 4-hydroxytamoxifen. ERα levels are quantified by immunofluorescence assay, in triplicate, 4-hr post compound treatment. Error bars are SEM. (**C**) Western blot analysis assessing the effect of GDC-0810 (100 nM), and fulvestrant (100 nM) (**D**) on ERα levels at 2, 4 or 6 hr of treatment, in the presence or absence of the 26S proteasome inhibitor MG132 (10 μM). (**E**) MCF7 cell viability assay comparing GDC-0810 activity to fulvestrant and 4OH-tamoxifen, in the absence of exogenous estradiol. Viable cells are presented as percent CellTiter-Glo luciferase activity relative to the vehicle control after 5 day compound incubation. Error bars represent standard deviation from the mean, from biological quadruplicates. (**F**) MCF7 cell viability assay comparing GDC-0810 activity to fulvestrant and 4OH-tamoxifen,

*Figure 1 continued on next page*

*Figure 1 continued*

in the presence of 0.1 nM estradiol. Viable cells are presented as percent CellTiter-Glo luciferase activity relative to the vehicle control after 5 day compound incubation.

DOI: https://doi.org/10.7554/eLife.15828.002

The following figure supplement is available for figure 1:

**Figure supplement 1.** MCF7 ERα In-Cell western assay correlates with MCF7 western blot for most ER binders.

DOI: https://doi.org/10.7554/eLife.15828.003

To assess the potential agonistic activity of GDC-0810 we evaluated the consequence of GDC-0810 treatment on a previously described SERM discriminatory gene set, likewise, in the absence of estrogen in MCF7 cells (*Wardell et al., 2013*). In line with earlier observations, 4OH-tamoxifen induces a strong up-regulation of a gene set that includes the estrogen-responsive genes, *AGR2* and *RASGRP*, while fulvestrant fails to do so (*Figure 2B*). GDC-0810 most closely resembled fulvestrant and GW7604 in this analysis, displaying little of the transcriptional agonist activity observed with the SERMs 4OH-tamoxifen, lasofoxifene, arzoxifene or endoxifene and often demonstrating inverse agonist activity on estradiol responsive genes (*Figure 2B*). These findings are consistent with ER-ChIP analysis demonstrating that GDC-0810 treatment, similar to fulvestrant, can reduce the levels of ERα associated with chromatin on ERα target genes (*Figure 2—figure supplement 1*).

Wardell et al. have demonstrated that fulvestrant and bazedoxifene can inhibit ER signaling and estrogen-dependent proliferation in the absence of ER degradation (*Wardell et al., 2011*, *Wardell et al., 2013*). This was demonstrated using an experimental system whereby ER was over-expressed in an attempt to saturate the degradation machinery, enabling an uncoupling of ER degradation from ER antagonism. Likewise, we sought to determine if GDC-0810 can inhibit ER signaling in the absence ER degradation, using a similar strategy (*Wardell et al., 2011*, *Wardell et al., 2013*). Over-expression of ER in doxycycline-inducible MCF7 cells resulted in the induction of the ER target gene, PR, consistent with ligand-independent activation of the pathway. Treating these cells with fulvestrant resulted in both depletion of ER, as well as suppression of PR. In the case of GDC-0810 treatment, PR was likewise suppressed, though ER levels remained high (*Figure 2C*). Cell viability assays demonstrated that both fulvestrant and GDC-0810 suppress proliferation of MCF7 cells over-expressing ER, with EC50s and IC50s similar to cells expressing endogenous levels of ER, despite the retention of high ER levels in the over-expressing cells treated with GDC-0810 (*Figure 2D,E*). These data imply that GDC-0810 has the ability to antagonize ER function independent from its degradation activity, consistent with the demonstration that GDC-0810 displaces the co-activator PGC1α in cell free biochemical assays, in which ER degradation does not occur (see below, Figure 6A).

**Table 1.** In vitro properties of GCD-0810.

| Compound | ER binding[*] | | Transcription[†] | | Cell viability[‡] | | ERα degradation[§] | |
| | ERα | ERβ | 3X ERE~LUC | | CellTiter-Glo | | In-Cell Western | |
| | $K_i$ [nM] | | $IC_{50}$ [nM] | $E_{max}$ [% E2] | $IC_{50}$ [nM] | $E_{max}$ [% E2] | $EC_{50}$ [nM] | $E_{max}$ [% Veh.] |
|---|---|---|---|---|---|---|---|---|
| GDC-0810 | 3.8 ± 1.6 | 3.7 ± 4.0 | 1.3 ± 0.8 | 6.1 ± 2.8 | 2.5 ± 2.1 | 24.6 ± 3.3 | 0.65 ± 0.50 | 15.3 ± 3.4 |
| 4-OH Tam | 2.2 ± 1.3 | 3.6 ± 1.7 | 6.7 ± 3.6 | 4.7 ± 2.9 | 0.53 ± 0.25 | 48.0 ± 4.7 | 0.14 ± 0.04 | 51.9 ± 2.7[#] |
| Fulvestrant | 13.1 ± 10.8 | 13.2 ± 7.6 | 0.3 ± 0.2 | 4.1 ± 2.6 | 0.56 ± 0.70 | 25.4 ± 3.7 | 0.39 ± 0.18 | 6.4 ± 2.0 |

[*] Binding affinities (Ki) of GDC-0810, 4-hydroxytamoxifen (4-OHT), and fulvestrant for ERα and ERβ. Shown are the mean and standard deviation of 3–4 experiments run in duplicate.

[†] ERα antagonist reporter assay. Results are the mean and standard deviation of 3 experiments.

[‡] Relative cell viability after 5 d incubation with compound. Shown are the mean and standard deviation of more than 50 assays run in triplicate.

[§] Relative ERα immunofluorescence activity in MCF7 In-Cell Western.

[#] The apparent reduction in ERα immunoreactivity is not reproduced in western blots.

DOI: https://doi.org/10.7554/eLife.15828.004

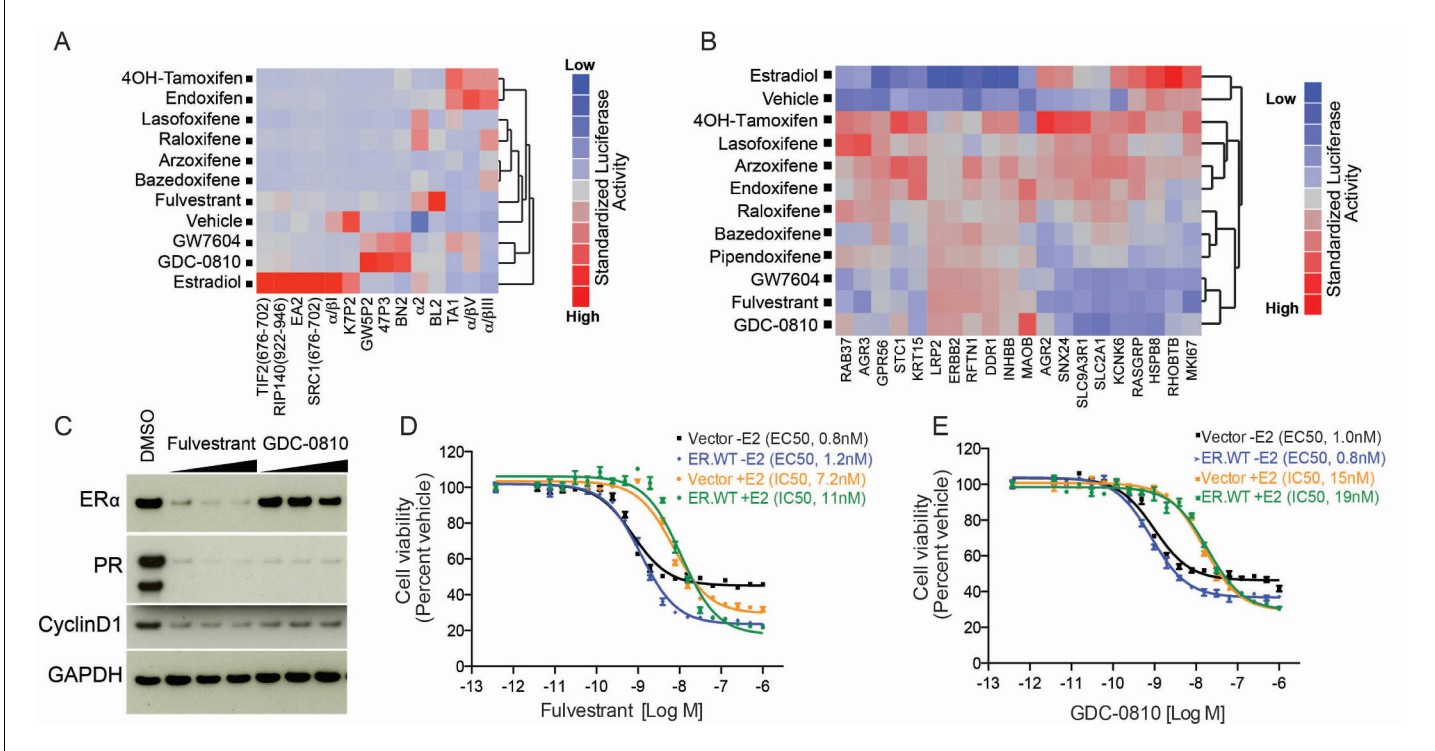

**Figure 2.** GDC-0810 promotes an ER confirmation distinct from 4OH-tamoxifen and fulvestrant. (**A**) ERα conformational profiling. A mammalian 2-hybrid assay was performed to monitor interaction of ERα with 14 conformation selective peptide probes. Luciferase signal was measured after 24 hr of compound treatment (1 μM for all compounds). Interaction profiles of ERα therapies, GDC-0810 and vehicle control from biological triplicates were analyzed by hierarchical clustering using the Ward algorithm and standardized data. (**B**) Transcriptional activity of benchmark ERα ligands in MCF7 cells, in the absence of exogenous estrogen. Transcriptional activity was monitored using a SERM discriminatory target gene set following 24 hr 1 μM ligand treatment. Data was log2 normalized followed by standardization and hierarchical clustering. (**C**) Doxycycline-inducible MCF7 cells were pre-cultured in estrogen-depleted medium with 100 ng/ml doxycycline for 2 weeks. Cells were then cultured in medium containing 10, 100 and 500 nM fulvestrant or GDC-0810 for 5 days, and the effect on levels of ER, PR and Cyclin D1 were assessed by Western Blot analysis. (**D**, **E**) Doxycycline-inducible MCF7 cells were pre-cultured in estrogen-depleted medium with 10 ng/ml doxycycline for 2 weeks before treatment. Cells were treated with a range of doses of fulvestrant or GDC-0810 for 7 days in estrogen-depleted medium with or without 0.1 nM E2. Cell viability was determined by CellTiter-Glo assay.
DOI: https://doi.org/10.7554/eLife.15828.005

The following figure supplement is available for figure 2:

**Figure supplement 1.** GDC-0810 does not promote the binding of ERα to known ER target sites.
DOI: https://doi.org/10.7554/eLife.15828.006

## GDC-0810 displays mild estrogenic activity in uterine models in vitro and in vivo

Since the activity of ER ligands can be tissue/cell-type dependent, we next evaluated the effects of GDC-0810 in the uterus, first making use of the Ishikawa endometrial cell line, in which alkaline phosphatase expression is controlled by ER (*Holinka et al., 1986*; *Littlefield et al., 1990*). In line with previous reports, 4OH-tamoxifen robustly increases alkaline phosphatase activity, while fulvestrant shows no such effect (*Figure 3A*). GDC-0810 modestly increases alkaline phosphatase activity, though plateaus at low concentrations, below the level stimulated by 4OH-tamoxifen (*Figure 3A*). To follow up on this observation we next assessed the consequences of GDC-0810 administration on the uterus of juvenile rats. Tamoxifen at 60 mg/kg, dosed orally, once a day for 3 days, increases uterine wet weight, though not to the extent driven by the administration of 0.1 mg/kg 17α estradiol over the same time period (*Figure 3B*). In contrast, Fulvestrant at 50 mg/kg acts as an inverse agonist, resulting in a significant decrease in uterine weight relative to the control group. GDC-0810 dosed at 0.1 or 10 mg/kg did not significantly alter uterine wet relative to the vehicle control (*Figure 3B*). Importantly though, histological analysis and quantification of endometrial cell height showed that, like estradiol and tamoxifen, GDC-0810 induced a cuboidal to columnar morphological

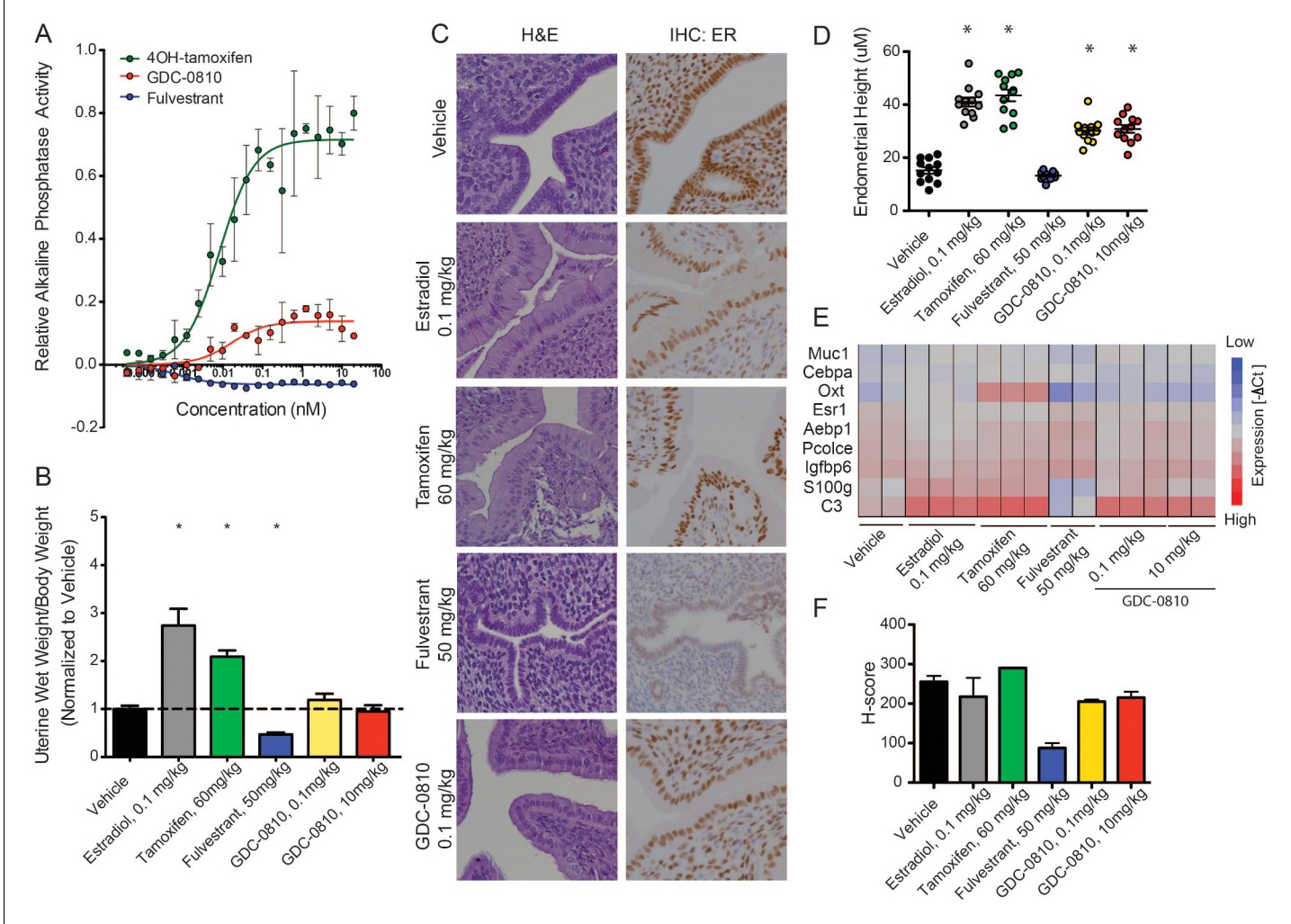

**Figure 3.** GDC-0810 displays mild estrogenic activity in vitro and in vivo. (A) Alkaline phosphatase activity in Ishikawa endometrial cells stimulated with increasing concentrations of either 4OH-tamoxifen, GDC-0810 or fulvestrant, in the absence of estrogen. (B) Uterine wet weight (UWW) measurements from juvenile rats treated with individual specified ER ligands; 17α-estradiol was dosed at 0.1 mg/kg as a positive control. (C) Haematoxylin and eosin (H&E) staining, and anti-ER IHC were performed on tissues dissected as in B. (D) Endometrial cell height was digitally measured from the basement membrane to the apical (luminal) surface, using a digitally scanned image at 20X magnification. Three digital measurements were taken from each section, and 2 mice per condition were scored. Results are displayed as the mean endometrial cell height from two animals ± standard error (n = 2). (E) Gene expression in the uterus, after treatment with indicated compounds, was assessed using a rat Fluidigm panel. (F) H-scores were used to quantify the ER IHC from (C). H-scores were estimated manually, incorporating both intensity and percentage of positive nuclei, using the following formula: (0 x % negative)+(1 x% weak)+(2 x% moderate)+(3 x% strong) * Denotes significance (p<0.05) compared to Vehicle in 1-Way ANOVA and Dunnett's Multiple Comparison Test.

DOI: https://doi.org/10.7554/eLife.15828.007

change in the epithelial cells, while the epithelia from fulvestrant-treated animals remained cuboidal (*Figure 3C,D*). In line with the observed morphological changes to the endometrial epithelium, gene expression analysis demonstrated that GDC-0810 induces ER target genes, including C3 (*Figure 3E*). These in vivo phenotypes were observed at a GDC-0810 dose of 0.1 mg/kg and were not significantly different at the 10 mg/kg dose, suggesting saturation of the effect at low doses. Together, these data demonstrate that GDC-0810 mildly stimulates ER signaling in the uterus, similar to the recently described SERD AZD9496 (*Weir et al., 2016*). Intriguingly, IHC analysis of ER in the uterus showed that GDC-0810 only modestly reduces ER levels in this context, similar to estradiol, while fulvestrant robustly depletes ER.

# GDC-0810 regresses tumors and suppresses estradiol uptake in tamoxifen-sensitive breast cancer tumor models

In mice, GDC-0810 exhibits low clearance (11 mL/min/kg) and 61% oral bioavailability (*Lai et al., 2015*). Importantly, GDC-0810 plasma concentrations increase proportionately with the applied dose and achieve an $AUC_{0-24}$ of 94.1 µg*hr/ml when dosed by oral gavage at 100 mg/kg/day (*Supplementary file 2A*). To determine the ability of GDC-0810 to inhibit E2 stimulated tumor growth in vivo, we treated MCF7 tumor bearing nu/nu mice orthotopically implanted with 0.36 mg/ 60 day release E2 pellets (Innovative Research) with GDC-0810, ranging from 1 to 100 mg/kg/day p.o. Fulvestrant was delivered by sub-cutaneous injection, with an initial loading dose of 50 mg/kg on days 1, 3 and 8, and subsequent dosing of 25 mg/kg twice per week to achieve exposures similar to those achieved in the clinic. In the MCF7 xenograft model, GDC-0810 displayed dose dependent efficacy (*Figure 4A*). The 100 mg/kg/day dose caused tumor regressions; an effect similar to withdrawal of estrogen pellets at the start of dosing. In contrast, fulvestrant, at a clinically relevant dose as well as at a considerably higher dose (200 mg/kg, 3 times per week), yielded only modest tumor growth inhibition (*Figure 4A* and *Figure 4—figure supplement 1A*). It is important to note that the restriction of the fulvestrant response to tumor growth inhibition, rather than stasis or tumor regression, was not a function of low exposure, as the plasma concentrations at the 200 mg/kg dose were on average 12–14 µg*hr/mL, approximately 30-fold above the clinical exposure of the fulvestrant 500 mg clinical regimen (*Ohno et al., 2010*; *Pritchard et al., 2010*). Gene expression analysis of harvested MCF7 tumors demonstrated that GDC-0810 (at 100 mg/kg/day, evaluated on day 28) robustly modulated ER target genes, including *PGR*, *c-MYC*, *AREG* and *MUC1* (*Figure 4B,C*; *Figure 4—figure supplement 1B*). Indeed, the gene expression changes induced by GDC-0810 are similar to, and in some cases even more pronounced than, those induced by withdrawal of the estrogen pellet at the beginning of the study, highlighting that GDC-0810 actively and efficiently attenuates ER signaling (*Figure 4B,C*; *Figure 4—figure supplement 1B*).

As an additional pharmacodynamic analysis in MCF7 xenografts, we next monitored tumor [18]F-fluoroestradiol (FES) uptake by positron emission tomography (PET), which has previously been used as a non-invasive, real-time measure of the activity of ER modulators [reviewed in (*Liao et al., 2016*)]. To determine the ability of GDC-0810 to effectively bind tumor-localized ER in vivo, FES-PET imaging was performed on mice bearing MCF7 xenograft tumors following a 7-day treatment with GDC-0810. Prior to GDC-0810 administration FES-PET signal was detectable in both the vehicle and pre-treatment groups (*Figure 4D*). However, when assayed after the seventh consecutive oral daily dose, GDC-0810 at 10 mg/kg/day and 100 mg/kg/day reduced FES uptake by 45 and 63 percent respectively (*Figure 4D,E*). At study termination on day 9, the GDC-0810 treatment groups displayed a dose dependent decrease in tumor volume and in ERα and PR immunohistochemical staining (*Figure 4—figure supplement 2*). While the difference in reduction of FES uptake between the 10 mg/kg/day and 100 mg/kg/day treatment groups did not reach statistical significance, each dose resulted in a statistically lower FES uptake compared to that in vehicle-treated animals. This would suggest that maximal GDC-0810-mediated ER antagonism, as a result of receptor occupancy and/or degradation, is achieved with a dose of ≥10 and ≤100 mg/kg/day in this model system.

We next evaluated the activity of GDC-0810 in patient derived xenograft (PDX) model, HCI-003. Like MCF7, HCI-003 tumors are ER+, E2 dependent and tamoxifen sensitive (*Figure 4—figure supplement 1C*). GDC-0810, dosed at 10 and 100 mg/kg/day in mice implanted with 1 mg E2 impregnated beeswax pellets, drove tumor stasis, while fulvestrant dosed at 30 times the clinical exposure induced tumor regression, approaching that observed in vehicle treated animals whose estradiol pellets were excised at start of the dosing period (*Figure 4F*). Consistent with GDC-0810 SERD activity in vitro, tumors collected after 43 days of dosing displayed reduced ERα target gene transcription and reduced ERα protein levels, as monitored by quantitative PCR and Western Blot, respectively (*Figure 4—figure supplement 3*). GDC-0810 also displayed anti-tumor activity in ZR75-1, an additional ER+ breast cancer xenograft model (*Figure 4—figure supplement 1D*). Importantly, GDC-0810 did not exhibit any anti-tumor activity in MDA-MB-231, an ER negative human breast cancer tumor model (*Figure 4—figure supplement 4*). Lack of efficacy in this ER-negative model is consistent with the GDC-0810 mechanism of action being selectively mediated through ER.

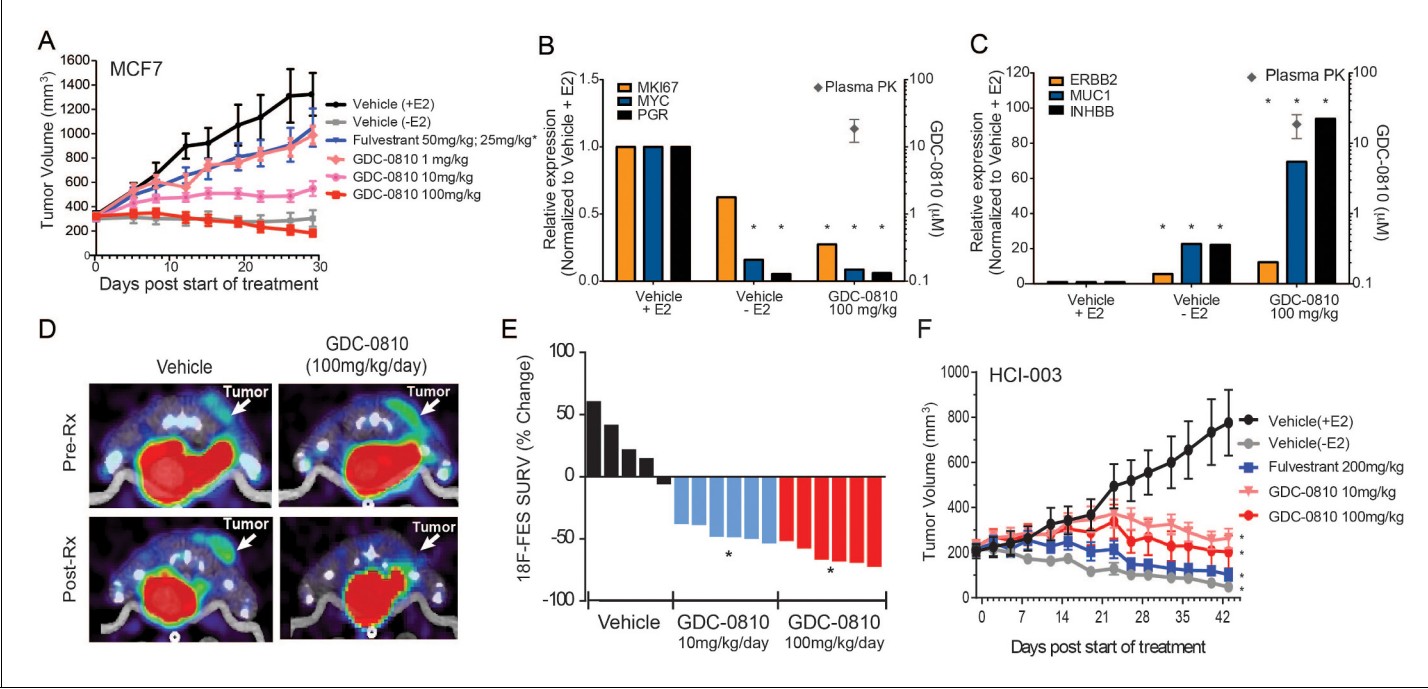

**Figure 4.** Antitumor activity and pharmacodynamic response of GDC-0810 in tamoxifen-sensitive breast cancer xenograft models. (**A**) Tamoxifen-sensitive MCF7 tumor bearing animals were dosed with vehicle, fulvestrant (50 mg/kg on days 1, 3, 8; then 25 mg/kg 2x/week, s.c.) or GDC-0810 (1, 10, 100 mg/kg/day, p.o.) for 28 days in the presence of 60-day release 0.36 mg 17β-estradiol pellets. (**B, C**) Gene expression analysis of tumors treated with 100 mg/kg GDC-0810, compared to tumors in the presence or absence of estrogen pellets. Tumors were harvested on day 28 of the study; this was a separate study from that shown in (**A**). GDC-0810 plasma concentration is also shown. *p<0.05, n = 3. See *Figure 4—figure supplement 1B* for an extended panel of genes. (**D**) Representative FES-PET images of MCF7 tumors in the right dorsum (arrow) of mice treated with vehicle or GDC-0810 (100 mg/kg). Images were taken 1–2 hr after the dosing on the seventh day of treatment. (**E**) Percent change in FES SUVR after 6 days of treatment. Each bar represents the mean percent change in 18F-labeled estradiol SUVR. Vehicle-treated mice exhibited an average increase of SUVR of 26.1% whereas mice treated with 10 mg/kg and 100 mg/kg exhibited a 45.2% and 63.3% reduction in SUVR, respectively, compared to baseline (*p<0.0001 vs. vehicle). (**F**) HCI-003 patient derived xenograft tumors were implanted in mice containing a 1 mg 17β-estradiol beeswax pellet. Tumor bearing animals were dosed with vehicle, fulvestrant (200 mg/kg, 3x/week, s.c.), GDC-0810 (10 or 100 mg/kg/day, p.o.) for 43 days. One vehicle treated group had the 17β-estradiol pellets removed at treatment start to assure growth dependence on estradiol.

DOI: https://doi.org/10.7554/eLife.15828.008

The following figure supplements are available for figure 4:

**Figure supplement 1.** MCF7, HCI-003 and ZR-75-1 breast cancer xenograft models.
DOI: https://doi.org/10.7554/eLife.15828.009

**Figure supplement 2.** Activity of GDC-0810 in MCF7 xenograft tumors analyzed by FES-PET.
DOI: https://doi.org/10.7554/eLife.15828.010

**Figure supplement 3.** Pharmacodynamic activity of GDC-0810 and fulvestrant in HCI-003 xenograft tumors.
DOI: https://doi.org/10.7554/eLife.15828.011

**Figure supplement 4.** GDC-0810 is not efficacious in MDA-MB-231, an ER negative human breast cancer tumor model.
DOI: https://doi.org/10.7554/eLife.15828.012

## GDC-0810 has anti-tumor activity in a tamoxifen-resistant MCF7 model

We previously generated a tamoxifen-resistant xenograft model, TamR1, by chronically treating MCF7 tumor bearing mice with tamoxifen until a resistant tumor emerged (*Lai et al., 2015*). Consistent with other reported xenograft models of tamoxifen resistance, treatment of the TamR1 xenografts with tamoxifen stimulated tumor growth, relative to the E2 pellet harboring vehicle controls [*Figure 5A*, and (*Connor et al., 2001*; *Gottardis and Jordan, 1988*; *Lai et al., 2015*)]. Our earlier work showed that treatment with GDC-0810 p.o. at 100 mg/kg/day induced tumor regressions in this model, while fulvestrant SC at 200 mg/kg (3x/week) exhibited only modest tumor growth inhibition (*Lai et al., 2015*). These experiments were conducted in mice with relatively high levels of circulating estradiol, with estradiol being delivered via 0.72 mg 60 day release E2 pellets that are

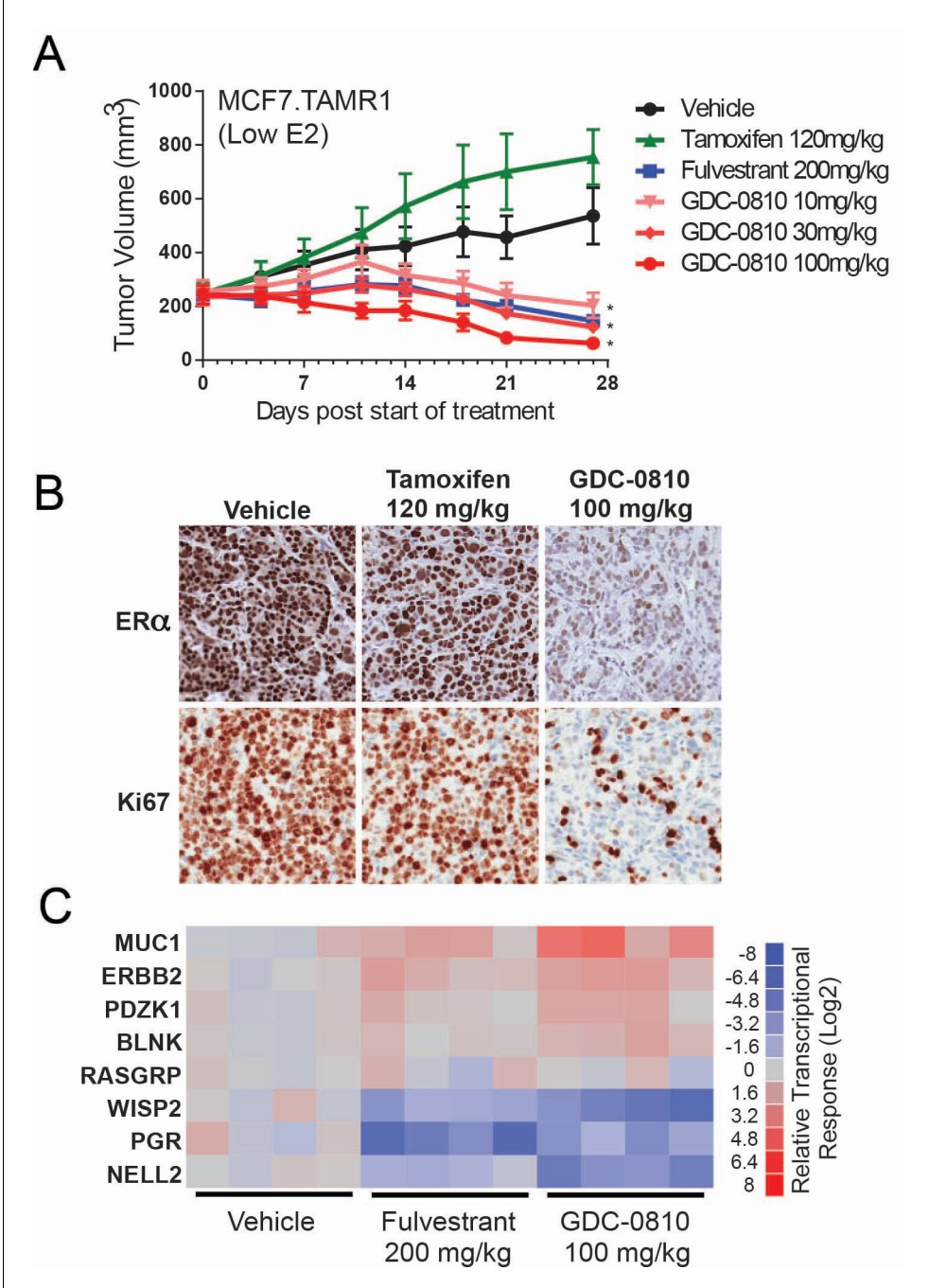

**Figure 5.** Antitumor activity of GDC-0810 in a tamoxifen-resistant breast cancer xenograft model. (**A**) Tamoxifen-resistant MCF7 tumors were implanted in animals supplemented with 60-day release 0.18 mg 17β-estradiol pellets. Tumor bearing animals were dosed with vehicle, tamoxifen (120 mg/kg/day p.o.), fulvestrant (200 mg/kg, 3x/week, s.c.) or GDC-0810 (10, 30 or 100 mg/kg/day, p.o.) for 27 days (**B**) Representative images of IHC for ERα and Ki67 in tamoxifen-resistant MCF7 xenograft tumors from (**A**) treated with vehicle, tamoxifen or GDC-0810. (**C**) Quantitative PCR analysis of ER-regulated genes in tamoxifen-resistant MCF7 xenograft tumors from (**A**), treated with vehicle, fulvestrant or GDC-0810. S.c. is sub-cutaneous dosing, and p.o. is per os (by mouth) oral gavage dosing. * Denotes significance ($p < 0.05$) compared to Vehicle in 1-Way ANOVA and Dunnett's Multiple Comparison Test.

DOI: https://doi.org/10.7554/eLife.15828.013

The following figure supplements are available for figure 5:

**Figure supplement 1.** Effects of estradiol pellet on ERα levels in MCF7-TamR1 xenograft tumors.

DOI: https://doi.org/10.7554/eLife.15828.014

*Figure 5 continued on next page*

*Figure 5 continued*

**Figure supplement 2.** Pharmacodynamic activity of GDC-0810 in MCF7-TamR1 xenograft tumors.
DOI: https://doi.org/10.7554/eLife.15828.015

reported by the manufacturer (Innovative Research) to deliver between 300 and 400 pg/ml estradiol. Under these conditions the high E2 reduces the steady state ERα protein levels, thus confounding the readout of ERα protein as a pharmacodynamic endpoint (*Figure 5—figure supplement 1*). Given this limitation of the 'high E2' models, the anti-tumor and pharmacodynamic response to GDC-0810 treatment was evaluated in the TamR1 model grown in mice implanted with pellets designed to release lower concentrations of estradiol (0.18 mg 60 day E2 pellets, reported to deliver 75–100 pg/mL estradiol). These pellets are sufficient to support tumor growth but only modestly reduce ERα levels (*Figure 5—figure supplement 1*).

In TamR1 tumors grown under these conditions of lower E2, GDC-0810 induced regression at all doses tested (*Figure 5A*). Fulvestrant, at the 200 mg/kg dose, where exposure is considerably higher than that achieved in the clinic, likewise resulted in regression of TamR1 tumors in animals harboring the 0.18 E2 pellet. Interestingly, the same dosing regimen of fulvestrant, showed only minor tumor growth inhibition in the presence of the 0.72 mg E2 pellet (*Lai et al., 2015*), suggesting that fulvestrant 200 mg (3x/week) does not effectively compete with the higher E2 levels elicited by the 0.72 mg E2 pellet. Similar to the in vitro observations, GDC-0810 treatment resulted in a dramatic reduction in ERα levels in the xenograft tumors by both immunohistochemistry and Western blot (*Figure 5B*; *Figure 5—figure supplement 2*). Consistent with the anti-tumor effect, GDC-0810 treated tumors display reduced Ki-67 staining (*Figure 5B* and *Figure 5—figure supplement 2*) and altered ER target gene transcription in a manner consistent with ER antagonism (*Figure 5C*).

## GDC-0810 antagonizes ERα ligand binding domain mutants in vitro and in vivo

Recently, a series of ESR1 mutations were identified in patients with metastatic ER+ breast cancer, who have progressed on aromatase inhibitors and other endocrine therapies (*Jeselsohn et al., 2014*; *Li et al., 2013*; *Merenbakh-Lamin et al., 2013*; *Robinson et al., 2013*; *Toy et al., 2013*). Based on overexpression studies, these mutations have been proposed to confer estrogen-independent activity to ERα and are thought to contribute to endocrine resistant disease (*Jeselsohn et al., 2014*; *Li et al., 2013*; *Merenbakh-Lamin et al., 2013*; *Robinson et al., 2013*; *Toy et al., 2013*). We focused on the two most commonly occurring mutations, ER.Y537S and ER.D538G, and addressed how these mutations might influence the activity of GDC-0810. In cell-free E2 competitive binding assays, GDC-0810 retains its ability to potently displace E2 from the ligand binding domain, albeit with a slightly increased IC50 (WT: 2.6 nM vs. ER.Y537S: 5.5 nM and ER.D538G: 5.4 nM) (*Figure 6A*). We next determined that GDC-0810 can compete the PGC1α co-activator peptide off the mutated ligand binding domain, implying that GDC-0810 is capable of driving an 'active' to 'inactive' conformational shift of mutant ER, though with a ~five–seven fold reduction in biochemical potency compared to wild-type ER (*Figure 6B*).

To further evaluate the activity of GDC-0810 in the mutant context, we engineered the Y537S mutation into the endogenous ESR1 allele in MCF7 cells, using CRISPR-Cas9 technology (*Figure 6—figure supplement 1A*). Twenty-four clonal lines were generated and their mutant allele frequencies, at the DNA level, were evaluated using mutant-specific droplet digital PCR (ddPCR) (*Figure 6—figure supplement 1B*). We selected 7 clones that contained the highest mutant allele frequency for further characterization. These clones were assessed for the expression of the mutant allele using ddPCR on cDNA, in cells grown in either standard media, or hormone-depleted charcoal stripped media. The mutant allele was expressed at ~33% of total mRNA, in all clones examined regardless of growth conditions. One of these clonal lines (clone 6) was selected for subsequent studies (*Figure 6—figure supplement 1C*). Despite expressing only 33% mutant RNA, ER.Y537S cells display a high level of estrogen-independent ER pathway activity, as demonstrated by high expression of the ER-target genes, TFF1, GREB1 and PGR, in the absence of estrogen, although these genes can be further stimulated with the addition of estrogen (*Figure 6C*). We then performed cell viability experiments, assessing the anti-proliferative activity of GDC-0810 in ER.WT and ER.Y537S cells. When the

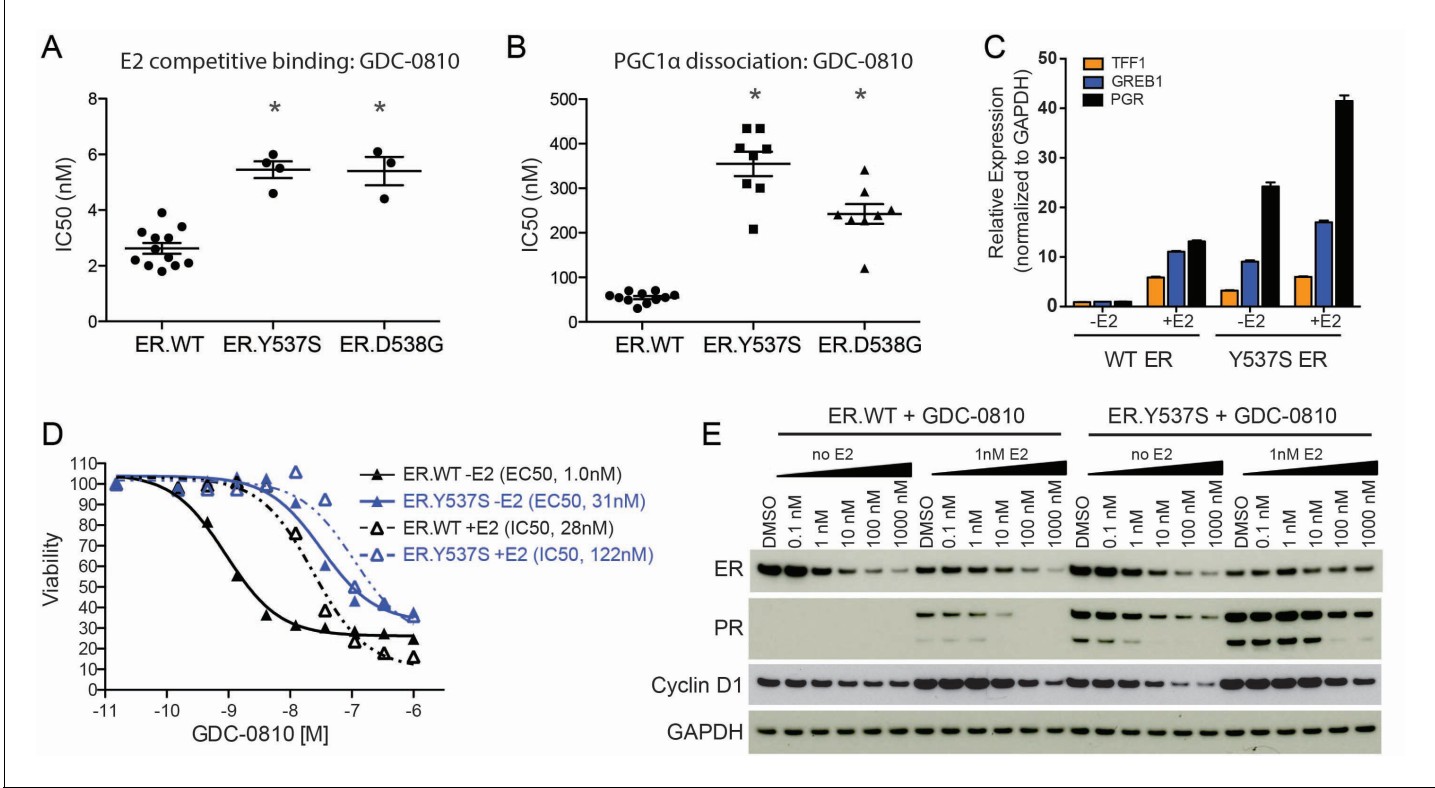

**Figure 6.** GDC-0810 antagonizes the estrogen-independent ER.Y537S mutant. (**A**) A cell free, FRET-based E2 competitive binding assay (E2 present at EC80) was used to determine the binding of GDC-0810 to ER.WT, ER.Y537S and ER.D538G ligand binding domains. Shown are the IC50 values calculated from multiple independent experiments, with mean and standard deviation indicated (**B**) A cell free, FRET-based PGC1α recruitment assay was used to evaluate the effect of GDC-0810 on ER:PGC1α interaction in the presence of agonist (EC80), using either purified wild-type or mutant ERα ligand binding domains (LBD). Shown are the IC50 values calculated from multiple independent experiments, with mean and standard deviation indicated. (**C**) Quantitative RT-PCR analysis of ER-regulated genes from CRISPR-Cas9 engineered ER.Y537S cells, in the absence and presence of estrogen, highlighting the E2 independent pathway activity of MCF7 cells expressing ER.Y537S. (**D**) Cell viability assays, measuring the effect of GDC-0810, were performed on MCF7 ER.WT (black lines) and ER.Y537S (blue lines) cells, in the absence (solid lines) and presence (dotted lines) of estrogen. (**E**) Western blot analysis evaluating levels of ERα, as well as PR and cyclin D1 as ER targets, in ER.WT and ER.Y537S cells. Cells were treated with GDC-0810 for 24 hr.

DOI: https://doi.org/10.7554/eLife.15828.016

The following figure supplement is available for figure 6:

**Figure supplement 1.** Characterization of MCF7 ER.Y573S knock-in cells.

DOI: https://doi.org/10.7554/eLife.15828.017

viability assay is conducted in the absence of E2, GDC-0810 displays a marked reduction in potency in the mutant versus wild-type setting, with 22 fold higher concentration being required to achieve 50% maximal inhibition of growth in the ER.Y537S versus ER.WT cells (*Figure 6D*, solid black line vs. solid blue line). In the presence of low E2 concentration (0.1 nM), this potency shift is less apparent, with only four fold increased compound concentration being required to achieve the same growth inhibition as in the ER.WT cells (*Figure 6D*, dashed black line vs. dashed blue line). The patterns are similar in the case of fulvestrant, with greater potency shifts occurring in the absence of estrogen versus the presence of estrogen (*Figure 6—figure supplement 1E*). The potency of 4OH-tamoxifen is marginally shifted in ER.Y537S cells relative to ER.WT cells (*Figure 6—figure supplement 1F*). The observed potency-shift for GDC-0810 in mutant cells in not restricted to clone 6, as this phenotype was seen in an additional 6 clones that were evaluated (*Figure 6—figure supplement 1D*).

We next compared the ability of GDC-0810 to reduce ER levels in ER.WT versus ER.Y537S cells, in the presence and absence of estrogen. Consistent with previous reports, estrogen itself reduces steady-state levels of ER, while simultaneously increasing PR expression (*Figure 6E*, lane 1 vs. lane 7). GDC-0810 reduces steady-state levels of ER protein, in both ER.WT and ER.Y537S cells,

demonstrating that the presence of ER.Y537S does not preclude GDC-0810-mediated ER-degradation, though due to a mixed population of ER.WT and ER.Y537S we cannot directly determine if GDC-0810 induced degradation of ER.Y537S is any more or less efficient than that of ER.WT (*Figure 6E*). The presence of ER.Y537S elevates PR protein levels in the absence of estrogen, consistent with the gene expression data, and this is further induced with the addition of estrogen. The absolute levels of PR achieved in mutant expressing cells are elevated compared to the levels seen in estrogen-treated wild-type ER cells, though this may be a feature of less than saturating/low E2 concentration. GDC-0810 reduces both estrogen-driven, and mutant-driven PR protein levels, athough residual PR is still detected in the mutant cells at 24 hr post compound addition (*Figure 6E*). Like PR, cyclin D1 levels are increased by both estrogen stimulation and by the presence of the ER.Y537S mutation; in both cases treatment with GDC-0810 blocks this increase, with cyclin D1 levels being comparable to un-stimulated cells. In contrast to the robust ER-depletion mediated by GDC-0810 and fulvestrant, tamoxifen does not deplete ER in either ER.WT or ER.Y537S cells, and robustly stabilizes ER in the presence of estrogen, in a dose dependent manner, while simultaneously decreasing levels of PR (*Figure 6—figure supplement 1F,H*). These data highlight the different consequences of the SERDs versus tamoxifen on ER levels, but demonstrate that GDC-0810, fulvestrant, and 4OH-tamoxifen are all mechanistically, and in principle, capable of suppressing ER signaling driven by the expression of ER.Y537S in vitro (*Figure 6—figure supplement 1F,H*).

As an additional ER mutant knock-in model, AAV (adeno-associated virus)-mediated homologous recombination was used to introduce the ER.D538G mutation into T47D cells, such that the mutation would be expressed under the endogenous promoter. T47D ER.D538G cells, expressing WT:D538G mRNA at a 1:1 ratio (data not shown) express endogenous levels of total ER, though very high levels of PR relative to WT cells, even in the absence of estrogen, implying robust estrogen-independent pathway activity (*Figure 7A*). Stimulation with estrogen results in depletion of ER in WT and D538G cells, as expected, though does not further induce PR in the mutant cells. In cell viability assays,

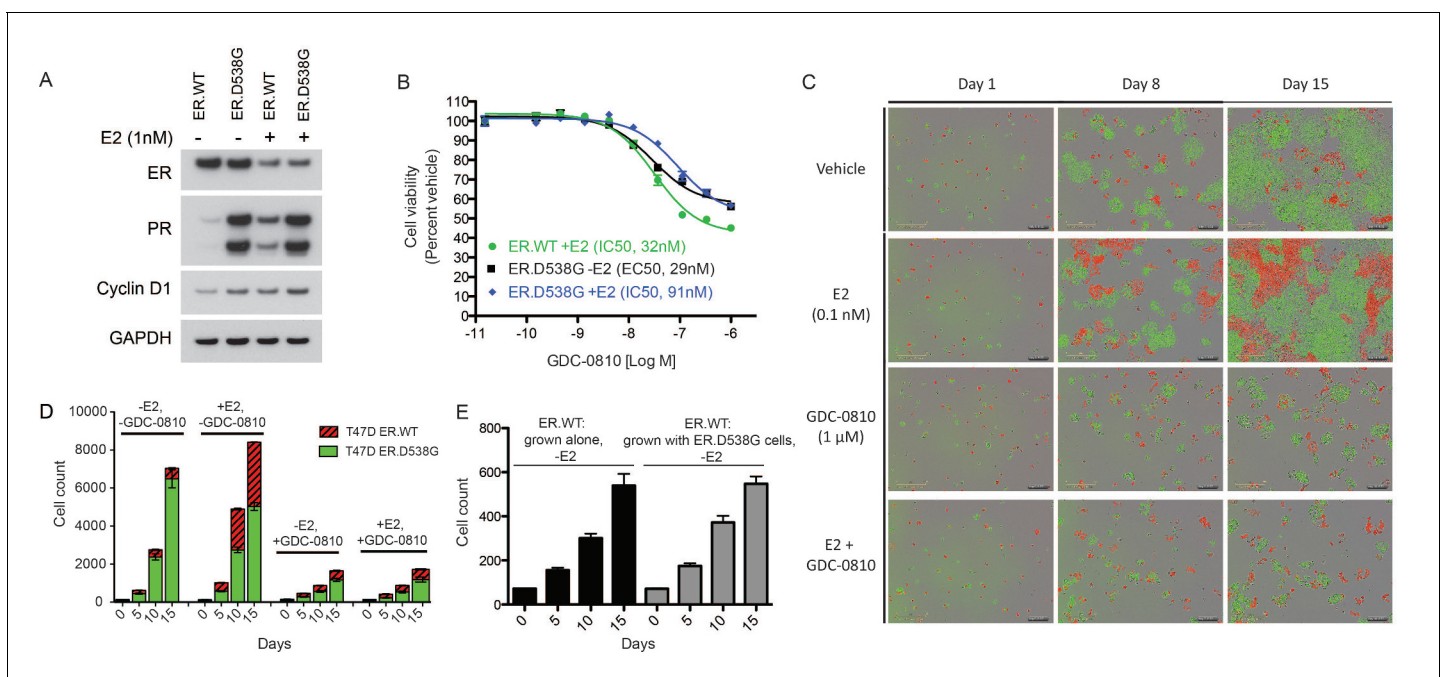

**Figure 7.** GDC-0810 exhibits activity in T47D ER.D538G knock-in cells. (**A**) Western blot analysis of T47D ER.WT and ER.D538G cells in the presence and absence of estrogen. (**B**) Cell viability assays of GDC-0810 activity in ER.WT and ER.D538G cells. (**C**) Representative images of T47D ER.WT (red) and ER.D538G (green) cells, grown under specified conditions. Wells, in which cells had been plates at a 1:1 ratio on day 1, were imaged on a regular basis to enable quantification of ER.WT/red vs. ER.D538G/green cells over time. (**D**) Quantification of cell competition experiments in which labeled T47D.WT cells (red), were mixed at a 1:1 ratio with labeled T47D.D538G cells (green), and grown in the presence and absence of estrogen and GDC-0810. 4 images per condition were quantified. (**E**) Cell counts for ER.WT cells grown alone or in the presence of ER.D538G cells.
DOI: https://doi.org/10.7554/eLife.15828.018

GDC-0810 suppresses the proliferation of both ER.WT cells and ER.D538G cells, with modest differences in potency (*Figure 7B*). We next fluorescently labeled the T47D ER.WT (with RFP) and ER.D538G cells (with GFP) and performed cell-cell competition assays, to further evaluate the conditions under which ER mutant cells have a competitive growth advantage relative to wild-type cells, and the effect of mutant ERα on response to GDC-0810. When mixed at a 1:1 ratio at day 0, and grown in the absence of estrogen, ER.D538G cells robustly outcompete ER.WT cells, with WT cells representing only 7.8% (± 1%) of the total population on day 15 (*Figure 7C,D*). In the presence of 0.1 nM estrogen however, much of this competitive advantage is lost, with WT cells representing 40% ± 0.2% of the total population at day 15 (*Figure 7C,D*). GDC-0810 attenuates both the estrogen-dependent growth of ER.WT cells, as well as estrogen-independent growth of the ER.D538G cells, though both cell types continue to proliferate at low levels in the presence of drug. In the case of the ER.WT cells, this rate of proliferation is equivalent to cells grown in the absence of estrogen, suggesting that ER-independent mechanisms may support low-level proliferation in these cells. These cell labeling/cell competition experiments further indicate that ER mutant cells do not confer a non-cell autonomous growth signal to ER.WT cells grown in the absence of estrogen, since ER.WT cells grow at the same rate, regardless of whether they are grown alone, or co-cultured with estrogen-independent ER.D538G cells (*Figure 7E*).

Though both the MCF7 ER.Y537S and T47D ER.D538G clones displayed constitutive ER activity in vitro, these cell lines did not form tumors when injected into nude mice in the absence of exogenous estradiol. Therefore, GDC-0810 was tested in a xenograft derived from MCF7 cells overexpressing amino-terminal HA-tagged ER.Y537S (HA-ER.Y537S), which formed rapidly growing tumors in ovariectomized mice without estradiol supplementation. Strikingly, GDC-0810 100 mg/kg/day p.o. induced tumor regressions in the HA-ER.Y537S overexpressing model, while tamoxifen 60 mg/kg/day p.o. and fulvestrant 200 mg/kg SC (3x/week) resulted in tumor stasis at exposures higher than those achieved in clinic (*Figure 8A*). As an additional in vivo ER mutant model, we utilized WHIM20, an ER.Y537S-expressing patient derived xenograft model (PDX) (*Li et al., 2013*). WHIM20 is heterozygous for the Y537S mutation, though the mutant allele represents close to 100% of the ESR1 transcript produced in these cells (*Figure 6—figure supplement 1C*). A pharmacodynamic (PD) experiment showed that GDC-0810, dosed orally at 100 mg/kg, and fulvestrant dosed intramuscularly at the high dose of 200 mg/kg for 4 days, both modulate a collection of known ER regulated targets, including robust suppression of RASGRP1 and AREG, as examples (*Figure 8B*). Western blot analysis, quantitative reverse phase protein array (RPPA) and IHC of ER and PR, in the same 4-day PD experiment, demonstrated that GDC-0810 and fulvestrant treatment resulted in modest decreases in ER and PR protein levels (*Figure 8C,D*). Importantly, in a separate efficacy study, GDC-0810 100 mg/kg/day treatment and tamoxifen 60 mg/kg/day induced tumor stasis, while fulvestrant at at a dose that roughly achieves plasma levels similar to those achieved with the 500-mg dose in humans (50 mg/kg days 1, 3 and 7, 25 mg/kg weekly thereafter; 0.015 mg/mL at trough on day 21, AUC ~0.36 µg*hr/mL) did not induce regression but resulted in tumor growth inhibition (*Figure 8E*, *Supplementary file 2B*).

## Discussion

Fulvestrant stands as proof of concept that the SERD class of ER modulators is an effective therapeutic modality to treat ER+ breast cancer. However, pre-clinical and clinical data indicate that fulvestrant's therapeutic action is limited by its poor pharmaceutical properties. GDC-0810 is a product of efforts to capitalize on the advantages associated with the SERD approach, while overcoming the limitations associated with fulvestrant. GDC-0810 is a non-steroidal, orally bioavailable dual function ER antagonist and down-regulator that induces potent, rapid proteasome-mediated ERα degradation in breast cancer cells. GDC-0810 binds ERα with low nanomolar potency, exhibits good oral bioavailability, and dose-dependent/linear pharmacokinetics in the mouse. When administered orally, once daily, at doses that deliver substantial plasma concentrations and are well-tolerated, GDC-0810 displays robust anti-tumor activity in cell line-based and patient derived xenograft models of breast cancer.

Though GDC-0810 clearly shares features with fulvestrant, such as the ability to down-regulate ER protein levels in breast cancer cells, the GDC-0810:ER complex interacts with a set of peptides that are distinct from those that interact with the fulvestrant:ER complex, suggesting that these

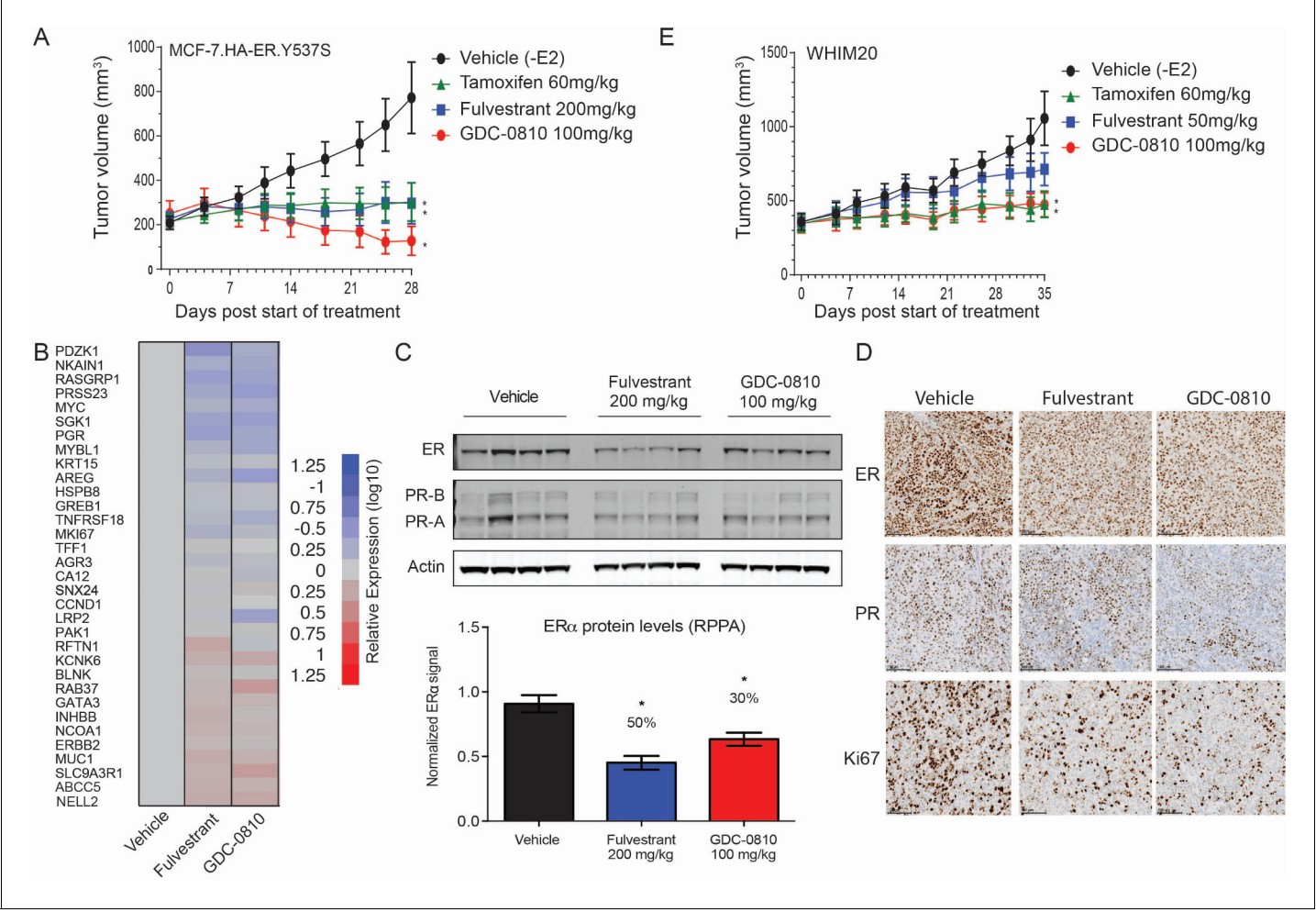

**Figure 8.** GDC-0810 antagonizes estrogen-independent ER.Y537S-expressing tumors in vivo. (**A**) MCF7 HA-ER.Y537S overexpressing tumors were implanted in animals without supplemental 17β-estradiol pellets. Tumor bearing animals were dosed with vehicle, tamoxifen (60 mg/kg/day p.o.), fulvestrant (200 mg/kg, 3x/week, s.c.) or GDC-0810 (100 mg/kg/day, p.o.) for 28 days. (**B**) Gene expression analysis of WHIM20 tumors after 4 days of dosing, harvested 8 hr after the final dose. (**C**) Western blot and quantification of ER levels by RPPA, assessing the effect of GDC-0810 and fulvestrant on ER and PR levels in the WHIM20 PDX model. (**D**) Representative IHC images of WHIM20 tumors treated with either vehicle, fulvestrant or GDC-0810 and probed using anti-ER, anti-PR, or anti-Ki67 antibodies. (**E**) WHIM20 tumors were implanted in mice without supplemental 17β-estradiol pellets. Upon reaching ~200 mm³, tumor bearing animals were dosed with vehicle, tamoxifen (60 mg/kg/day p.o.), fulvestrant (50 mg/kg, days 1, 3 and 8, 25 mg/kg 1x/week thereafter, s.c.) or GDC-0810 (100 mg/kg/day, p.o.) for 35 days. * Denotes significance (p<0.05) compared to Vehicle in 1-Way ANOVA and Dunnett's Multiple Comparison Test.

DOI: https://doi.org/10.7554/eLife.15828.019

compounds promote distinct ER conformations, and may therefore have a different mechanism by which they induce ER turnover. The observations that 1) saturation of the GDC-0810 and the fulvestrant ER degradation machinery occur differentially in our doxycycline-inducible ER-over-expressing MCF7 cell line (*Figure 2C*) and 2) that fulvestrant, but not GDC-0810, drives degradation of ER in the rat uterus (*Figure 3C,F*), further supports the notion that these compounds have distinct mechanisms by which they induce ER degradation, and potentially require differing co-factors or distinct sets of E3 ligases. The observation that activation of the ER pathway occurs concomitantly with lack of robust ER degradation upon GDC-0810 treatment in the rat uterus suggests that although ER degradation is not absolutely required for direct ER antagonism, it may be associated with preventing ER agonism, at least in this context.

The behavior of GDC-0810 in the conformational profiling assay is itself intriguing; specifically the robust recruitment of the GW5P2 and 47P3 peptides to the ER:GDC-0810 complex, which are not

recruited by any of the already marketed SERM/SERD:ER complexes. GW5P2 was experimentally derived from efforts to identify unique, selective conformation-sensing peptides using a random 10-residue peptide phage display library in the presence of GW5638 or its metabolite GW7604 (*Iannone et al., 2004*). 47P3 was subsequently generated as an attempt to make a second generation, higher affinity, more specific peptide for the GW7604:ER complex. Notably, the binding of the GW5P2 and 47P3 peptides to various ER:ligand complexes was highly correlated with the presence of an acrylic or carboxylic acid head group in ligand structures (*Iannone et al., 2004*). The binding of GW5P2 and 47P3 to the ER:GDC-0810 complex thus likely reflects the presence of this acrylic acid side chain in GDC-0810. It was speculated that these selective peptides may be affinity-optimized versions of naturally occurring co-repressor peptides which are yet to be discovered (*Iannone et al., 2004*). Future identification of such co-repressors may provide a foothold in our efforts to understand the molecular mechanism by which GDC-0810 antagonizes ER function.

FES-PET imaging, which we used as a real-time pharmacodynamic biomarker in the MCF7 xenograft model, has previously been used to monitor the activity of endocrine agents in the clinic (*Linden et al., 2011*, *Linden et al., 2006*). A recent analysis demonstrated that reductions in FES-PET signal following fulvestrant treatment were both highly variable and incomplete (<75% reduction in SUV) (*van Kruchten et al., 2015*). Furthermore, the lack of robust inhibition of FES-PET signal was associated with lack of clinical benefit (*van Kruchten et al., 2015*). FES-PET imaging is likewise being utilized in clinical studies of GDC-0810. In the Phase I dose escalation study in postmenopausal women with ER+ locally advanced or metastatic breast cancer (clinicaltrials.gov NCT01823835), complete/near complete (>90%) suppression of FES uptake was reported to be observed in 90% of patients with FES-PET scans, including 5 patients with known ESR1 mutations, providing early clinical evidence of robust ER engagement by GDC-0810 [(*Dickler et al., 2015*), and manuscript in preparation]. Since FES-PET signal does not distinguish competitive antagonism from ER depletion; a thorough molecular evaluation of pre- and post- treatment biopsies will be required to better understand how ER antagonism coupled with degradation impacts ER signaling and ultimately clinical response.

The landmark discoveries of highly prevalent, recurrent ESR1 mutations in patients that have failed multiple lines of endocrine therapy highlights the sustained reliance of ER+ breast cancer on ER function, and further emphasizes the need for next generation ER-targeted therapeutics. We generated ER.Y537S and ER.D538G knock-in mutant cell lines to model the two most commonly occurring clinical mutations. Despite expressing 50% or less of the mutant allele, these cells display robust estrogen-independent ER signaling and estrogen-independent cell proliferation. Cell competition assays further highlight that much of the competitive advantage of the mutant cells occurs in the absence of estrogen, mimicking AI-treatment, and that this competitive advantage is lost both in the presence of estrogen and in the presence of GDC-0810. These data are consistent with recent retrospective clinical studies showing that ESR1 mutations in plasma DNA predict relative resistance to AIs but not to fulvestrant (*Fribbens et al., 2016*; *Schiavon et al., 2015*; *Spoerke et al., 2016*). In patients, these mutations were frequently sub-clonal, and the question was raised as to whether estrogen-independent ER mutant cells might confer estrogen-independence to neighboring ER wild-type cells. Based on our observations from the T47D model, this appears not to be the case, as ER.WT cells exhibit the same growth rate, regardless of whether they are co-cultured with ER.D538G-expressing cells or grown alone.

GDC-0810 maintains its ability to induce ER degradation, to suppress ER signaling, and to inhibit proliferation in cell lines that express mutant ER, though a higher drug concentration is needed. Structural studies and molecular dynamics simulations of ER mutants have demonstrated that the Y537S and D538G substitutions shift ER, and particularly helix 12 (H12), into a conformation that resembles an active/liganded ER wild-type conformation, providing a molecular explanation for the constitutive activity of these mutations (*Nettles et al., 2008*; *Toy et al., 2013*). A recent biophysical analysis of these mutants has provided an important extension of those findings, demonstrating that this 'pre-organized agonist state' restricts access to the ligand binding domain, which influences binding of both estrogen and tamoxifen, and presumably other ligands, including GDC-0810, which is consistent with what we show here (*Fanning et al., 2016*). Fanning et al. further demonstrate that tamoxifen fails to fully restore H12 as well as the H11-12 loop back to the wild-type antagonist state, providing a further explanation for reduced tamoxifen potency in the mutant context. They hypothesize that highly potent, next generation SERDs, including GDC-0810, may have an increased ability

to destabilize H12 and thereby drive efficacy in the ER mutant setting. Our observations that despite the somewhat reduced potency in vitro, GDC-0810 displays robust activity in mutant models in vivo, are in line with this hypothesis, although we acknowledge that tamoxifen likewise exhibits robust in vivo efficacy in the WHIM20 ER.Y537S model. The data demonstrating in vivo activity for GDC-0810 in ER mutant models is consistent with: 1) the in vivo exposures of GDC-0810 being sufficient to overcome the potency shifts caused by ER.Y537S in vitro, and 2) the level of ER pathway suppression driven by GDC-0810 in the mutant context being sufficient for in vivo anti-tumor activity. How these pre-clinical studies will relate to clinical observations is an ongoing question; the activity of GDC-0810 in ER mutant versus wild-type patients is being carefully monitored in clinical studies.

The clinical activity of fulvestrant supports ER degradation as a desirable feature of ER therapeutics targeting ER+ breast cancer. However, the question still remains as to how much advantage SERDs will have relative to ER ligands that function as potent antagonists in the absence of ER degradation. Indeed, our data shows that despite differences in activity in vitro, tamoxifen, like GDC-0810 is efficacious in ER.WT and mutant models in vivo, and it could be argued that the superior activity of GDC-0810 could in some cases be due to its potent antagonistic effects coupled together with good bioavailability and PK properties, rather than its ability to lower ER protein levels. Along these lines, we show that GDC-0810's ER degradation activity can be uncoupled from its antagonistic activity in MCF7 cells. A recent study likewise highlighted the activity of bazedoxifene, a SERD/SERM hybrid that displays only modest ER degradation activity relative to fulvestrant and GDC-0810, as having robust efficacy against ER.WT and mutant models both in vitro and in vivo (*Wardell et al., 2015*). It stands to reason that in many cases, antagonism of the ER signaling pathway should be sufficient to drive anti-tumor activity. We argue that the benefit of SERDs vs. non-degrader antagonists will be most apparent in those contexts where ER becomes reactivated in the presence of drug, such as is observed in the TamR1 xenograft, in which tamoxifen behaves as a strong agonist. Since the relatively limited number of pre-clinical in vivo models fails to recapitulate the multiple potential mechanisms of ER-dependent resistance, nor the profound heterogeneity that exists across ER+ breast cancer patients, we feel that this is a question that will ultimately be addressed in the clinic, as we gain additional experience with GDC-0810 and other SERDs.

## Materials and methods

### Cell culture and reagents

MCF7 (ATCC), T47D (ATCC), ZR-75-1 (ATCC) cells were maintained in RPMI 1640. Ishikawa (Sigma), HEK293T (ATCC) and MDA-MB-231 (ATCC) cells were maintained in DMEM. CV1 (ATCC) and HEPG2 (ATCC) cells were maintained in MEM. All medium was supplemented with 10% fetal bovine serum (FBS) (Hyclone), 1 mM sodium pyruvate and 1X non-essential amino acids unless otherwise indicated. Unless indicated, tissue culture supplements and medium were purchased from Mediatech or Invitrogen. GDC-0810, arzoxifene and bazedoxifene were synthesized at Seragon Pharmaceuticals or Genentech. Fulvestrant and raloxifene were purchased from Waterstone Technology LLC. 4OH-tamoxifen, endoxifen, and 17β-estradiol were purchased from Sigma Aldrich. Lasofoxifene was purchased from Toronto Research Chemicals.

### Cell line authentication/quality control is conducted by Genentech's centralized cell repository and is conducted as follows

Short tandem repeat (STR) profiling

STR profiles are determined for each line using the Promega PowerPlex 16 System. This is performed once and compared to external STR profiles of cell lines (when available) to determine cell line ancestry.

### Loci analyzed

Detection of sixteen loci (fifteen STR loci and Amelogenin for gender identification), including D3S1358, TH01, D21S11, D18S51, Penta E, D5S818, D13S317, D7S820, D16S539, CSF1PO, Penta D, AMEL, vWA, D8S1179 and TPOX

## SNP fingerprinting

SNP profiles are performed each time new stocks are expanded for cryopreservation. Cell line identity is verified by high-throughput SNP profiling using Fluidigm multiplexed assays. SNPs were selected based on minor allele frequency and presence on commercial genotyping platforms. SNP profiles are compared to SNP calls from available internal and external data (when available) to determine or confirm ancestry. In cases where data is unavailable or cell line ancestry is questionable, DNA or cell lines are re-purchased to perform profiling to confirm cell line ancestry.

## SNPs analyzed

rs11746396, rs16928965, rs2172614, rs10050093, rs10828176, rs16888998, rs16999576, rs1912640, rs2355988, rs3125842, rs10018359, rs10410468, rs10834627, rs11083145, rs11100847, rs11638893, rs12537, rs1956898, rs2069492, rs10740186, rs12486048, rs13032222, rs1635191, rs17174920, rs2590442, rs2714679, rs2928432, rs2999156, rs10461909, rs11180435, rs1784232, rs3783412, rs10885378, rs1726254, rs2391691, rs3739422, rs10108245, rs1425916, rs1325922, rs1709795, rs1934395, rs2280916, rs2563263, rs10755578, rs1529192, rs2927899, rs2848745, rs10977980

## Mycoplasma testing

All stocks are tested for mycoplasma prior to and after cells are cryopreserved. Two methods are used to avoid false positive/negative results: Lonza Mycoalert and Stratagene Mycosensor.

Cell growth rates and morphology are also monitored for any batch-to-batch changes.

## In-cell-western assay

Trypsinized MCF7 cells were washed twice in in phenol-red-free RPMI 1640 (supplemented with 5% CSS, sodium pyruvate, and non-essential amino acids), adjusted to a concentration of 200,000 cells in the same RPMI culture medium, and dispensed in 16 µL aliquots into flat clear bottom TC-Treated 384 Well plates (Corning, NY, USA). Cells were incubated for 72 hr, after which ligand diluted in RPMI culture medium was added in a 16 µL volume. After 4 hr' incubation, 16 µL of 30% neutral buffered formalin was added directly to the 32 µL cell culture. The fixed cells were permeabilized with PBS containing 0.1% Triton X-100, washed with PBS containing 0.1% Tween, blocked with Odyssey Blocking Buffer (LI-COR), incubated with Rabbit anti-ERα antibody (SP-1; Thermo Scientific), washed, incubated with IRDye 800 CW goat anti rabbit secondary antibody and Draq5 DNA stain. Plates were washed and ERα and DNA levels were quantitated using a LI-COR Odyssey infrared imaging system. ER levels were normalized to DNA. Percent ERα was defined as normalized ERα sample/normalized ERα untreated cells × 100.

## Western blot

Proteins from cell lysates were separated electrophoretically using NuPAGE 4–12% Bis Tris Gels (Life technologies) in MOPS buffer (Life Technologies). Gels were then electroblotted onto Nitrocellulose Pre-Cut Blotting Membranes (Life Technologies), blocked with LI-COR blocking buffer (LI-COR), incubated with Rabbit anti-ERα antibody (SP-1; Thermo Scientific) and mouse Anti-α-Tubulin antibody (DM1A, Sigma-Aldrich). Membranes were washed with 0.1% tween–20 in PBS. Membranes were then incubated with IRDye 800 CW goat anti rabbit and IRDye 680 CW goat anti mouse secondary antibodies (LI-COR), washed, and scanned using a LI-COR Odyssey infrared imaging system. ERα and α-tubulin levels were quantitated using a LI-COR Odyssey infrared imaging system. ER levels were normalized to α-tubulin. Percent ERα was defined as normalized ERα sample/normalized ERα untreated cells × 100.

## Viability assays

Trypsinized MCF7 cells were adjusted to a concentration of 40,000 cells per mL in RPMI 1640 (supplemented with 10% FBS, sodium pyruvate, and non-essential amino acids), and dispensed in 16 µL aliquots into 384 well plates. Cells were incubated for 24 hr, after which ligand diluted in RPMI culture medium was added in a 16 µL volume. After 5 days' incubation, 16 µL of CellTiter-Glo Luminescent Cell Viability Assay (Promega, Madison, WI) was added and Relative Luminescence Units (RLUs) measured. Cell viability data is presented as relative luciferase activity defined as RLU sample/RLU untreated × 100. For the comparison of MCF7 ER.WT and ER.Y537S cells, WT cells were pre-

conditioned in charcoal stripped FBS for 3 days, and mutant cells are consistently grown in such conditions. Additionally, these particular assays were performed over 7 days.

## Engineering of MCF7 and T47D cells

### Doxycycline-inducible ER over-expressing MCF7 cells

lentiviral supernatants were generated by co-transient transfection of a lentivirus plasmid pIN-CUCER20 encoding ERα, the plasmid expressing the vesicular stomatitis virus (VSV-G) envelope glycoprotein and packaging plasmid delta 8.9 in HEK293T cells using Lipofectamine (Invitrogen). MCF-7 target cells were then transduced with lentiviral supernatants and infected cells were selected for neomycin (G418) resistance. Cells were characterized for doxycycline-inducible ER expression using western blot analysis.

### MCF7 CRISPR ER.Y537S knock-in cells

MCF7 cells expressing the ER.Y537S variant were generated using CRISPR-Cas9 gene editing. The donor vector, containing the mutation of interest in the 3' homology arm, and the gRNA, designed to cleave upstream of exon 10, along with Cas9 expression vector, were transfected into MCF7 cells. Pools of MCF7 cells in which donor DNA had been incorporated were obtained after positive (puromycin) and counter (ganciclovir) selection. Clones derived from single cells were then subject to PCR and Sanger sequencing, and subsequently to ddPCR to determine the allelic frequencies. Positive clones were infected with Ad-CMV-iCre to excise the selection cassette, which was followed by GFP-negative cell sorting.

### HA-ER.Y537S stable over-expressing MCF7 cells

This cell line was generated by sub-cloning ESR1 wild-type and mutant cDNAs containing an amino terminal hemagglutinin tag into pCDH-EF1-MCS-(PGK-Puro) (System Biosciences). The resulting plasmids, in addition to an empty vector negative control, were subsequently co-transfected into HEK293T cells with the pPACKH1 packaging plasmid mix (System Biosciences) according to the manufacturer's protocol. After transfection, lentiviral particles were purified from the cell medium, and used to transduce MCF7 cells. Stable cell lines were selected by growth in RPMI containing 10% FBS plus 1 μG/mL puromycin. Following selection, expression of HA-tagged mutant ERα was confirmed by western blot using the 6E2 mouse monoclonal anti-HA antibody (Cell Signaling).

### T47D ER.D538G knock-in cell line

T47D cells expressing ER.D538G were generated by Horizon, using rAAV-mediated homologous recombination. The cell line was sequence confirmed and established to express 50% mutant ESR1. Cells were fluorescently labeled using IncuCyte NucLight Lentivirus reagents according to manufacturers instructions.

## ERα co-activator peptide (PGC1α) antagonist assay

Test compounds were serially diluted in DMSO then diluted in TR-FRET Co-regulator Buffer E (Life Technologies PV4540) and 2 ml was transferred to a 1536-well (Aurora Biotechnologies MaKO 1536 Black Plate, #00028905) using a Biomek FX. A Beckman Coulter Bioraptr Dispenser was used to dispense: 2 ml per well of '3x ERα solution': 22 nM ERα (human estrogen receptor alpha, GST-tagged ESR1 ligand binding domain, spanning residues S282-V595, with either wild-type sequence or containing the mutations: Y537S or D538G) in TR-FRET Co-regulator Buffer E containing 7.5 mM dithiothreitol (DTT); and 2 ml of '3x assay mix' containing: 750 nM Fluorescein-PGC1α peptide sequence: EAEEPSLLKKLLLAPANTQ; (Life Technologies PV4421), 12 nM Estradiol, 15 nM LanthaScreen Tb-anti-GST antibody (Life Technologies, A15113) in TR-FRET Co-regulator Buffer E (with 7.5 mM DTT). Plates were centrifuged and incubated for 2 hr at room temperature. TR-FRET measurements were made using a Perkin Elmer EnVision Fluorescence Reader using Excitation: 340 nm and Emission: 495 nm and 520 nm. Percentage inhibition values were calculated relative to no compound (dimethylsulfoxide only) controls and a 'no ERα controls'. Curve fitting and IC50 calculations were carried out using Genedata Screener software.

## Juvenile rat uterus assays

### Uterine wet weight

At 21 days of age, Sprague Dawley rats were weaned, randomized into groups (n = 5) and treated once a day for 4 days with compounds of interest at indicated doses. Rats were euthanized 4 hr after the final dose. Uteri were then dissected, weighed and fixed for histological examination. A small portion of tissue was saved prior to fixation for mRNA isolation and gene expression analysis.

### Quantification of endometrial cell heights

H&E stained sections from bilateral representative uterine horn cross-sections were examined from two animals treated with the compounds shown. Endometrial cell height was digitally measured from the basement membrane to the apical (luminal) surface (orange line), using a digitally scanned image at 20X magnification. Obliquely cut areas were avoided. Three digital measurements were taken from each section (n = 6). Results are displayed as the mean endometrial cell height in two animals ± standard error (n = 2).

### ER IHC

Estrogen receptor was detected using the mouse monoclonal anti-ER (clone 1D5, Dako), incubated for 60 min at RT at a dilution of 0.5 µg/ml, and visualized after secondary rabbit anti-mouse antibody binding with Powervision DAB (Leica Powersystems). H-scores were estimated manually, incorporating both intensity and percentage of positive nuclei, using the following formula: (0 x% negative)+(1 x% weak)+(2 x% moderate)+(3 x% strong)

## Transcriptional reporter assays

Specific details of the ERα conformational profiling assay, and MCF7, ERα mutant, ERβ, PR-A, PR-B, MR and GR transcriptional reporter assays are presented in the Appendix.

## Nuclear receptor binding assays

Competitive radioligand binding assays were performed using either purified protein (for ERα, ERβ and GR) or cell extracts (for AR) as described in the Appendix.

## RNA isolation, quantitative PCR assays and ddPCR assays

Transcriptional activity of ER modulated genes from in vitro and in vivo samples, and ddPCR assays are described in the Appendix.

## Chromatin immunoprecipitation assay

ChIP assays were performed as described in the Appendix.

## In vivo pharmacology

Animal studies were conducted in accordance with the Guide for the Care and Use of Laboratory Animals, National Academy Press (2006), conforming to California State legal and ethical practices and approved by the Institutional Animal Care and Use Committee (IACUC, Seragon and/or Genentech).

Depending on the tumor line and experimental paradigm, animals were ovariectomized and/or implanted with estradiol impregnated pellets to stimulate tumor growth. Tumor fragments or cell suspensions were implanted subcutaneously on the right flank. Specific experimental and treatment paradigms are listed in the Supplemental information. Tumors were measured in two dimensions twice weekly. Volume in $mm^3$ was calculated by the formula: volume = length $\times$ (width$^2$)/2. When tumors reached an average size of roughly 150 – 350 $mm^3$, animals were randomized into groups and treatment was started. Animals were sacrificed after the final dose and tumors were excised, cut into approximately 30 mg fragments and flash frozen for pharmacodynamic analysis. Additional tumor fragments for immunohistochemistry were placed in 10% Neutral Buffered Formalin for 24 hr, and transferred to 70% ethanol until processing. 5 µm sections were labeled for ERα (Abcam#16660) or Ki67 (Abcam#16667) and stained with HRP/DAB detection kit (Abcam #64261). Details of the

FES/PET-CT imaging of the MCF7 xenograft tumors are provided in the Supplementary Methods and materials.

## Acknowledgements

We would like to thank Gina Wang, Jae Chung, Jun Liang, Tom O'Brien, Jim Kiefer and Melissa Junttila for helpful discussions and contributions during the preparation of the manuscript.

## Additional information

### Competing interests

James D Joseph, Beatrice Darimont, Steven P Govek, Andiliy G Lai, Mehmet Kahraman, Dan Brigham, John Sensintaffar, Nhin Lu, Gang Shao, Jing Qian, Kate Grillot, Michael Moon, Rene Prudente, Eric Bischoff, Kyoung-Jin Lee, Karensa L Douglas, Jackaline D Julien, Johnny Y Nagasawa, Anna Aparicio, Josh Kaufman, Richard A Heyman, Peter J Rix, Nicholas D Smith, Jeffrey H Hager: Shareholders of Seragon. Wei Zhou, Alfonso Arrazate, Amy Young, Ellen Ingalla, Kimberly Walter, Robert A Blake, Jim Nonomiya, Zhengyu Guan, Lorna Kategaya, Benjamin Haley, Jennifer M Giltnane, Ingrid E Wertz, Mark R Lackner, Michelle A Nannini, Deepak Sampath, Lori Friedman, Ciara Metcalfe: Employed by Genentech and own shares. Carlos L Arteaga: was a consultant for Genentech in 2015 and 2016. The consulting activities were not related to the manuscript. CLA is a member of the Komen Foundation's Scientific Advisory Board. The other authors declare that no competing interests exist.

### Funding

| Funder | Grant reference number | Author |
| --- | --- | --- |
| Vanderbilt-Ingram Cancer Center | P30 CA68485 | Luis Schwarz<br>Henry Charles Manning<br>Mohammed Noor Tantawy<br>Carlos L Arteaga |
| Susan G. Komen for the Cure | SAC100013 | Luis Schwarz<br>Henry Charles Manning<br>Mohammed Noor Tantawy<br>Carlos L Arteaga |
| Breast Cancer Specialized Program of Research Excellence | P50 CA098131 | Carlos L Arteaga |
| Vanderbilt-Ingram Cancer Center | Support Grant P30 CA68485 | Carlos L Arteaga |

The funders had no role in study design, data collection and interpretation, or the decision to submit the work for publication.

### Author contributions

James D Joseph, Conception and design, Acquisition of data, Analysis and interpretation of data, Drafting or revising the article; Beatrice Darimont, Conception and design, Acquisition of data, Analysis and interpretation of data; Wei Zhou, Alfonso Arrazate, Amy Young, Ellen Ingalla, Kimberly Walter, Robert A Blake, Jim Nonomiya, Zhengyu Guan, Lorna Kategaya, Steven P Govek, Andiliy G Lai, Mehmet Kahraman, Dan Brigham, John Sensintaffar, Nhin Lu, Gang Shao, Jing Qian, Kate Grillot, Michael Moon, Rene Prudente, Eric Bischoff, Kyoung-Jin Lee, Celine Bonnefous, Karensa L Douglas, Jackaline D Julien, Johnny Y Nagasawa, Anna Aparicio, Josh Kaufman, Jennifer M Giltnane, Luis Schwarz, Henry Charles Manning, Mohammed Noor Tantawy, Acquisition of data, Analysis and interpretation of data; Benjamin Haley, Ingrid E Wertz, Mark R Lackner, Michelle A Nannini, Deepak Sampath, Carlos L Arteaga, Richard A Heyman, Peter J Rix, Nicholas D Smith, Conception and design, Analysis and interpretation of data; Lori Friedman, Ciara Metcalfe, Jeffrey H Hager, Conception and design, Analysis and interpretation of data, Drafting or revising the article

Author ORCIDs
Ciara Metcalfe http://orcid.org/0000-0001-7233-661X

## Ethics

Animal experimentation: Animal studies were conducted in accordance with the Guide for the Care and Use of Laboratory Animals, National Academy Press (2006), conforming to California State legal and ethical practices and approved by the Institutional Animal Care and Use Committee (IACUC, Seragon and/or Genentech).

## Decision letter and Author response

Decision letter https://doi.org/10.7554/eLife.15828.025
Author response https://doi.org/10.7554/eLife.15828.026

## Additional files

### Supplementary files

• Supplementary file 1. Supplementary data tables related to the specificity for GDC-0810 in binding and activation of ER relative to other nuclear hormone receptors. (**A**) Radioligand binding assay (**B**) GDC-0810 nuclear hormone receptor reporter activity; agonist mode (**C**) GDC-0810 nuclear hormone receptor reporter activity; antagonist mode.
DOI: https://doi.org/10.7554/eLife.15828.021

• Supplementary file 2. GDC-0810 and fulvestrant mouse pharmacokinetic data. (**A**) GDC-0810 mouse pharmacokinetics (**B**) Fulvestrant plasma concentrations.
DOI: https://doi.org/10.7554/eLife.15828.022

• Supplementary file 3. Primer sequences. (**A**) Transcriptional Real-time PCR Oligonucleotide Sequence (**B**) ER-ChIP Real-time PCR Oligonucleotide Sequence.
DOI: https://doi.org/10.7554/eLife.15828.023

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

# Appendix 1

DOI: https://doi.org/10.7554/eLife.15828.020

## DNA and plasmids

3X ERE TK-luciferase, ERα-VP16, MH100-Luciferase, pCMX-Gal4 SRC1, and pCDNA-ERα plasmids were provided by Dr. David Mangelsdorf. MMTV-Luciferase was provided by Dr. Charles Sawyers. Gal4-ER interacting peptide expression constructs were generated by subcloning annealed oligonucleotides encoding the following peptides TIF2 (676–702). RIP140 (922–946), EA2, α/βI, K7P2, GW5P2, 47P3, BN2, α2, BL2, TA1, α/βV, and α/βIII into pCMX-Gal4 (provided by Dr. David Mangelsdorf). 2X PRE-Luciferase was created by subcloning 2 consensus PRE sequences (GTAGCTAGAACATCCTGTACA) into pGL4.26 (Promega). pCMX PRa was generated by subcloning full-length PRa into pCMX. pCDNA PRb was generated by cloning full-length human PRb into pCDNA. 6R MR and pCDNA3 GR were provided by Dr. Beatrice Darimont. pRL-CMV was purchased from Promega. pCMV-hspCas9-nickase-H1-gRNA and pCDH-EF1-MCS-(PGK-Puro) were purchased from System Biosciences.

## Binding assays

### ERα and ERβ competitive binding assay

Full length estrogen receptor α or β (Life Technologies) was diluted in TEGM buffer (10 mM Tris-HCl, 1 mM EDTA, 10% glycerol, 1 mM beta-mecaptoethanol, 10 mM Sodium Molybdate, pH 7.2, 0.04% NP-40) to a final concentration of 2.5 nM or 2 nM, respectively, combined with compound, 1.5 nM $^3$H Estradiol [2,4,6,7-3H(N)]- (Perkin Elmer), and incubated overnight at 4°C. The reaction was bound under vacuum to a Unifilter-96 GF/C filter plate had been pre-wetted with 0.5% poly-ethyleneimine. The plate was washed with wash buffer (40 mM Tris pH7.5, 100 mM KCl, 1 mM EDTA, 1 mM EGTA) and dried. Microscint-20 (Perkin Elmer) liquid scintillation counter cocktail was added to the filter plate, which was sealed with TopSeal-A adhesive sealing tape (Perkin Elmer), and the reactions were counted in a TopCount NXT Microplate Scintillation and Luminescence Counter (Perkin Elmer). Counts per minute were obtained and percent binding was determined (cpm ER + compound/cpm ER-compound × 100) and plotted against the compound concentration in order to obtain the IC50 for each compound using GraphPad Prism. Ki was calculated according to Cheng-Prusoff (*Cheng and Prusoff, 1973*) as Ki = IC50/(1 + ([3H-E2]/Kd)), [3H-E2] = 1.5 nM.

### AR competitive binding assay

MDA-MB-453 cells (ATCC) were cultured in RPMI 1640 containing 20 mM Hepes, 4 mM l-Glutamine, 10 µg/mL human Insulin, 10% FBS and 20 µg/mL Gentamicin. After reaching 90% confluence, cells were harvested, resuspended in TEGM (10 mM TrisHCl pH 7.2, 1 mM EDTA, 10% glycerol, 1 mM β-mercaptoethanol, 10 mM Sodium Molybdate), and frozen in liquid nitrogen in 10 mL aliquots containing $2 \times 10^7$ cells/mL. Binding reactions (60 µL) were carried out in 96-well plates in TEGM, and typically contained 24 µL cell lysate, 0.6 nM $^3$H -R1881 (Perkin Elmer), $3 \times 10^{-10}$ to $3 \times 10^{-5}$ M of the respective competitive ligand, and $10^{-11}$ to $10^{-6}$ M of the control ligand (R1881). Reactions were incubated at 4°C overnight. Bound and unbound ligands were separated by ultrafiltration using a Unifilter-96 GF/C filter plate (Perkin Elmer). Bound $^3$H-R1881 was eluted in 30 µL/well Microscint-20 and quantified using a scintillation counter (Top Count).

### GR competitive binding assay

The binding affinities of SRN-927 and comparators to GR were determined using competitive radiometric binding experiments. Reactions were conducted in 96-well plates in TEGM (10 mM Tris-HCl pH 7.2, 1 mM EDTA, 10% glycerol, 1 mM β-mercaptoethanol, 10 mM sodium

molybdate). Each reaction (60 µL) contained 2.0 nM GR, 3.0 nM $^3$H-dexamethasone (Perkin Elmer), and $3 \times 10^{-10}$ to $3 \times 10^{-5}$ M of the respective competitive ligand (SRN-927, 4-hydroxytamoxifen, or fulvestrant), or $10^{-11}$ to $10^{-6}$ M of the control ligand (dexamethasone). All reactions were set up in duplicates and incubated at 4°C overnight. Receptor-bound $^3$H-DEX was isolated by ultrafiltration using a Unifilter-96 GF/C filter plate (Perkin Elmer), eluted in 30 µL Microscint-20 and quantified using a scintillation counter (Top Count).

### ERα conformational profiling assay

HEK-293T cells were maintained in DMEM supplemented with 10% FBS. Conformational profiling assays were performed by seeding 100 µL of cells at a density of 250,000 cells/mL into 96-well cell culture plates in phenol red free DMEM, supplemented with 5% charcoal-stripped serum. After the cells were allowed to attach overnight, cells were transiently transfected using Lipofectin (Life Technologies). Triplicate transfections were performed using 150 ng MH100-Luciferase (Gal-4 responsive luciferase reporter), 130 ng pVP16-ERα (VP16-ERα expression vector), 150 ng pCMX-Gal4-peptide (Gal4-ERα interacting peptide expression vector), 50 ng pRL-CMV (normalization vector) and 0.6 µL lipofectin. Transfected cells were incubated overnight then treated with ligand.

Compounds, at a single saturating dose of 1 µM, were diluted in phenol red free DMEM supplemented with charcoal stripped serum to 12 µM and 10 µL was added to the cells. Following 24 hr incubation, the medium was removed and the cells were lysed in 40 µL of lysis buffer (25 mM Tris Phosphate, 2 mM CDTA, 10% Glycerol, 0.5% Triton X-100, and 2 mM DTT).

Firefly luciferase activity was measured immediately following the addition of 40 µL luciferase buffer (20 mM tricine, 0.1 mM EDTA, 1.07 mM $(MgCO_3)_4$ $Mg(OH)_2 \cdot 5H_2O$, 2.67 mM $MgSO_4$, 33.3 mM DTT, 270 µM Coenzyme A, 470 µM luciferin, 530 µM ATP). Renilla luciferase was measured following the addition of 40 µL coelenterazine buffer (1.1 M NaCl, 2.2 mM $Na_2EDTA$, 0.22 M $K_3PO_4$ (pH 5.1), 0.44 mg/mL BSA, 1.3 mM $NaN_3$, 1.43 µM coelenterazine, final pH adjusted to 5.0). Luminescence was measured immediately following reagent addition using either Perkin Elmer EnVision 2103 Multilabel Reader. Luminescence activity was normalized by dividing firefly luciferase activity by Renilla luciferase activity. Data was standardized and hierarchical clustered with the Ward algorithm using JMP8 (SAS).

## Reporter assays

### MCF7 Transcriptional reporter assay

ERα transcriptional reporter activity was monitored in MCF7 cells without transfection of supplemental ER. MCF7 cells were maintained in RPMI 1640 supplemented with 10% FBS. Transcriptional assays were performed by seeding 100 µL of cells at a density of 250,000 cells/mL into 96-well cell culture plates in phenol red free RPMI 1640 supplemented with 5% charcoal stripped serum and allowed to attach overnight. Cells were transiently transfected using Lipofectin (Life Technologies). Triplicate transfections were performed using 210 ng 3X ERE-TK-Luc (reporter vector), 30 ng pRL-CMV (normalization vector) and 0.6 µL lipofectin. Transfected cells were incubated overnight then treated with ligand.

For ER agonist assays, the compounds were serially diluted and 50 µL of compound plus RPMI 1640 supplemented with charcoal stripped serum was added to the cells. For ER antagonist assays, the compounds were serially diluted and 50 µL of compound with RPMI plus 17β-estradiol supplemented with charcoal stripped serum were added to the cells. The final 17β-estradiol concentration used in the antagonist assays was 0.1 nM. Following 24 hr incubation the medium was removed and the cells were lysed in 40 µL of lysis buffer (25 mM Tris Phosphate, 2 mM CDTA, 10% Glycerol, 0.5% Triton X-100, 2 mM DTT).

Firefly luciferase activity was measured immediately following the addition of 40 µL luciferase buffer (20 mM tricine, 0.1 mM EDTA, 1.07 mM $(MgCO_3)_4$ $Mg(OH)_2 \cdot 5H_2O$, 2.67 mM $MgSO_4$, 33.3 mM DTT, 270 µM Coenzyme A, 470 µM luciferin, 530 µM ATP). Renilla luciferase was measured following the addition of 40 µL coelenterazine buffer (1.1 M NaCl, 2.2 mM $Na_2EDTA$, 0.22 M $K_3PO_4$ (pH 5.1), 0.44 mg/mL BSA, 1.3 mM $NaN_3$, 1.43 µM coelenterazine,

final pH adjusted to 5.0). Relative luminescence was measured immediately following reagent addition using either Perkin Elmer EnVision 2103 Multilabel Reader or Molecular Devices Analyst GT Multimode Reader.

## ERβ transcriptional reporter assay

HEK-293T cells were maintained in DMEM supplemented with 10% FBS. Transcriptional assays were performed by seeding 100 µL of cells at a density of 250,000 cells/mL into 96-well cell culture plates in phenol red free DMEM supplemented with 5% charcoal stripped serum and allowed to attach overnight. Cells were transiently transfected using Lipofectin (Life Technologies). Triplicate transfections were performed using 25 ng pCMX ER-β, 210 ng 3X ERE-TK-Luc (reporter vector), 30 ng pRL-CMV (normalization vector) and 0.6 µL lipofectin. Transfected cells were incubated overnight then treated with ligand.

For ER agonist assays, the compounds were serially diluted and 50 µL of compound plus DMEM supplemented with charcoal stripped serum was added to the cells. For ER antagonist assays, the compounds were serially diluted and 50 µL of compound with DMEM plus 17β-estradiol supplemented with charcoal stripped serum were added to the cells. The final 17β-estradiol concentration used in the antagonist assays was 1 nM. Following 24 hr incubation the medium was removed and the cells were lysed in 40 µL of lysis buffer (25 mM Tris Phosphate, 2 mM CDTA, 10% Glycerol, 0.5% Triton X-100, 2 mM DTT).

Firefly luciferase activity was measured immediately following the addition of 40 µL luciferase buffer (20 mM tricine, 0.1 mM EDTA, 1.07 mM $(MgCO_3)_4$ $Mg(OH)_2 \cdot 5H_2O$, 2.67 mM $MgSO_4$, 33.3 mM DTT, 270 µM Coenzyme A, 470 µM luciferin, 530 µM ATP). Renilla luciferase was measured following the addition of 40 µL coelenterazine buffer (1.1 M NaCl, 2.2 mM $Na_2EDTA$, 0.22 M $K_3PO_4$ (pH 5.1), 0.44 mg/mL BSA, 1.3 mM $NaN_3$, 1.43 µM coelenterazine, final pH adjusted to 5.0). Relative luminescence was measured immediately following reagent addition using either Perkin Elmer EnVision 2103 Multilabel Reader or Molecular Devices Analyst GT Multimode Reader.

## PR-A, PR-B, and MR transcriptional reporter assays

HEPG2 cells were maintained in RPMI 1640 supplemented with 10% FBS. Transcriptional assays were performed by seeding 100 µL of cells at a density of 250,000 cells/mL into 96-well cell culture plates in RPMI 1640 supplemented with 5% charcoal-stripped serum and allowed to attach overnight. Cells were transiently transfected using Lipofectin (Life Technologies). Triplicate transfections were performed using 418 ng reporter vector (2X PRE-Luciferase for PR-A and PR-B, MMTV-Luciferase for MR), 50 ng CMVpRL (normalization vector), 10 ng NHR expression vector (CMX-PR-A, pCDNA-PR-B or 6R-MR), and 0.7 µL lipofectin. Transfected cells were incubated overnight then treated with ligand. For agonist assays, the compounds were serially diluted and 50 µL of compound plus RPMI 1640 supplemented with charcoal-stripped serum was added to the cells. For antagonist assays, the compounds were serially diluted and 50 µL of compound with RPMI plus 3 nM of the appropriate agonist (R-5020 for PR and aldosterone for MR) supplemented with charcoal-stripped serum were added to the cells. The final agonist concentration used in the antagonist assays was 1 nM. Following 24 hr incubation, the medium was removed and the cells were lysed in 40 µL of lysis buffer (25 mM Tris Phosphate, 2 mM CDTA, 10% Glycerol, 0.5% Triton X-100, 2 mM DTT). Firefly luciferase activity was measured immediately following the addition of 40 µL luciferase buffer (20 mM tricine, 0.1 mM EDTA, 1.07 mM $(MgCO_3)_4$ $Mg(OH)_2 \cdot 5H_2O$, 2.67 mM $MgSO_4$, 33.3 mM DTT, 270 µM Coenzyme A, 470 µM luciferin, 530 µM ATP). Renilla luciferase was measured following the addition of 40 µL coelenterazine buffer (1.1 M NaCl, 2.2 mM $Na_2EDTA$, 0.22 M $K_3PO_4$ (pH 5.1), 0.44 mg/mL BSA, 1.3 mM $NaN_3$, 1.43 µM coelenterazine, final pH adjusted to 5.0). Relative luminescence was measured immediately following reagent addition using either the Perkin Elmer EnVision 2103 Multilabel Reader or Molecular Devices Analyst GT Multimode Reader.

## GR transcriptional reporter assays

CV1 cells were maintained in MEM supplemented with 10% FBS. Transcriptional assays were performed by seeding 12 mL of cells at a density of 250,000 cells/mL into a T-75 cell culture flask in MEM supplemented with 5% charcoal-stripped serum and allowed to attach overnight. Cells were transiently transfected using FuGENE 6 (Roche Applied Science). Bulk transfections were performed using 1250 ng reporter vector (MMTV-Luciferase), 225 ng pRL-CMV (normalization vector), 25 ng pCDNA3-GR vector, and 75 μL FuGENE 6. Transfected cells were incubated overnight then split into four 96-well tissue culture plates and incubated for 4 hr followed by compound addition. For agonist assays, the compounds were serially diluted and 50 μL of compound plus MEM supplemented with charcoal-stripped serum was added to the cells. For antagonist assays, the compounds were serially diluted and 50 μL of compound with MEM plus 30 nM dexamethasone supplemented with charcoal-stripped serum were added to the cells. The final agonist concentration used in the antagonist assays was 10 nM. Following 24 hr incubation, the medium was removed and the cells were lysed in 40 μL of lysis buffer (25 mM Tris Phosphate, 2 mM CDTA, 10% Glycerol, 0.5% Triton X-100, 2 mM DTT). Firefly luciferase activity was measured immediately following the addition of 40 μL luciferase buffer (20 mM tricine, 0.1 mM EDTA, 1.07 mM $(MgCo_3)_4$ $Mg(OH)_2 \cdot 5H_2O$, 2.67 mM $MgSO_4$, 33.3 mM DTT, 270 μM Coenzyme A, 470 μM luciferin, 530 μM ATP). Renilla luciferase was measured following the addition of 40 μL coelenterazine buffer (1.1 M NaCl, 2.2 mM $Na_2EDTA$, 0.22 M $K_3PO_4$ (pH 5.1), 0.44 mg/mL BSA, 1.3 mM $NaN_3$, 1.43 μM coelenterazine, final pH adjusted to 5.0). Relative luminescence was measured immediately following reagent addition using either Perkin Elmer EnVision 2103 Multilabel Reader or Molecular Devices Analyst GT Multimode Reader.

## RNA isolation and real-time PCR (RT-PCR)

For most in vivo studies, approximately 30 mg of tumor was homogenized in 1 mL Trizol and RNA was prepared according to manufacturers instructions. For the pharmacodynamic/gene expression experiment described in *Figure 3B,C* and Figure 3—figure supplement 1B, flash frozen tumors were pulverized to powder by mechanical force (Covaris). Approximately 30 milligrams of tumor powder was lysed by addition of Buffer RLT (Qiagen) and homogenized by mechanical disruption (SPEX SamplePrep) and centrifugation through a QIAShredder column (Qiagen). RNA was subsequently isolated from homogenized lysates using the MagMAX-96 Total RNA Isolation Kit (Thermo Fisher Scientific), according to the manufacturer's protocol. For in vitro studies, cells were plated in phenol-free RPMI containing 5% charcoal dextran treated FBS. After 3 days the cells were treated with compound for approximately 20 hr. RNA isolation was performed as above. RNA (1 μg) was reverse-transcribed using the iScript cDNA synthesis kit (BIO-RAD, Hercules, CA). RT-PCR was performed using the Applied Biosystems 7900HT instrument and SYBR Green PCR Master Mix (Applied Biosystems, Foster City, CA). GAPDH expression was used to normalize all real-time data. Expression data was log2 standardized and graphed in cell plot using JMP8 software (SAS). Real-time PCR primer sequences are listed in *Supplementary file 3A*. For the gene expression analysis in *Figure 3B,C* and Figure 3 - figure supplement, total RNA was reverse-transcribed to cDNA and pre-amplified with a pool of gene specific primers. Following amplification, samples were diluted one to four with Tris-EDTA (TE) buffer and qPCR was conducted on Fluidigm 96.96 Dynamic Arrays using the BioMark HD system according to the manufacturer's protocol. Samples were assayed in duplicate and gene expression data were analyzed using RealTime StatMiner qPCR data analysis software. Gene expression data were normalized to the mean of three reference genes (*PPIA, SDHA, UBC*) and relative expression values were calculated using the Delta Delta Ct method. Bar charts were graphed using Prism 6.0 (GraphPad) and heat maps were generated using JMP11 (SAS).

## Chromatin immunoprecipitation assay

Cells were plated in 150 mm dishes ($7 \times 10^6$ cells in 20 mL) in RPMI 1640 supplemented with 10% CSS for 48 hr. Cells were treated with GDC-0810 and 4-hydroxytamoxifen at 1 µM, Fulvestrant at 100 nM, and E2 at 10 nM for 45 min or 4 hr. Following ligand treatment, formaldehyde was added to the media to a final concentration of 1%, incubated for 10 min and quenched with glycine (125 mM final concentration) for 5 min. Cells were washed 3X with PBS containing 1X Halt Protease & Phosphatase Single-Use Inhibitor Cocktail (1X PI, Thermo Scientific), pelleted, lysed in 1 mL RIPA buffer (50 mM Tris pH7.5, 0.15 M NaCl, 1% NP-40, 0.5% Na-deoxycholate, 0.05% SDS, 1 mM EDTA, 1X PI) and sonicated until the average DNA size fragment was ≈500 bp. The sonicated cross-linked chromatin was diluted into 3.3 mL RIPA and precleared with 100 mL 50% protein A/G agarose slurry (SC-2003, Santa Cruz Biotechnology) containing 200 mg per mL sonicated salmon sperm DNA and 500 mg per mL of BSA. One mL of precleared chromatin was then immunoprecipitated with 15 µg anti-ER (HC-20, Santa Cruz Biotechnology) or normal rabbit IgG (SC-2027, Santa Cruz Biotechnology), for 2 hr at 4°C and 100 mL of a 50% slurry of protein A/G agarose beads were added and incubated overnight at 4°C. Beads were washed 2 times sequentially in low-salt buffer (50 mM HEPES pH 7.8, 140 mM NaCl, 1 mM EDTA, 1% Triton X-100, 0.1% Na-deoxycholate, 0.1% SDS), high-salt buffer (same as low-salt with 500 mM NaCl), LiCl buffer (20 mM Tris pH 8.0, 1 mM EDTA, 250 mM LiCl, 0.5% NP-40, 0.5% Na-deoxycholate), and TE buffer (50 mM Tris pH 8.0, 1 mM EDTA). All washing steps were perfomed in the presence of 1X PI. Protein-DNA complexes were eluted in 225 mL Elution buffer (50 mM Tris pH 8.0, 1 mM EDTA, 1% SDS) at 65°C twice for 15 min. Eluted protein-DNA complexes were reverse cross-linked in the presence of NaCl overnight at 65°C and further treated with EDTA and proteinase K at 42°C for 1 hr. The DNA fragments were purified in 10 mM Tris pH 8.5 using the QIAquick PCR purification kit (Qiagen), diluted and analyzed by real-time PCR using iTaq SYBR Green Supermix with ROX (Bio-Rad). The samples were amplified on the ABI 7900HT instrument. Oligonucleotide primer sequences are listed in *Supplementary file 3B*.

## Droplet digital PCR (ddPCR)

Droplet digital PCR probe assays containing 5'HEX-labeled (WT) or 5'FAM-labeled (Mutant) probes were designed to detect the Y537S (1980 A > C), and D538G (1983 A > G) *ESR1* mutations within the ER ligand-binding domain. *ESR1* Y537S assay sequences were: Forward, GTACAGCATGAAGTGCAA; Reverse, GGGCGTCCAGCATC; WT Probe, AGCAGGTCATAGAGGGG; Y537S Mutant Probe, AGCAGGTCAGAGAGGG and D538G assay sequences were: Forward, TACAGCATGAAGTGCAAG; Reverse, TGGGCGTCCAGCA; WT Probe, CCCCTCTATGACCTGCT; D538G Mutant Probe, TCTATGGCCTGCTGCT. All assays were performed on the QX200 Droplet Digital PCR System (BioRad) according to the manufacturer's protocol. Each assay run incorporated *ESR1* WT and mutant oligo controls and a human scramble WT genomic DNA control (Roche), which were used to determine fluorescence amplitude thresholds for positive and negative mutation calls. Samples were assessed for the presence and allele frequencies of *ESR1* mutations using 20 ng input cell line genomic DNA. For assessment of ESR1 mutant RNA allele frequencies, 1 µg total cell line RNA was reverse-transcribed using the High Capacity cDNA Reverse Transcription Kit (Applied Biosystems), and 10 ng input cDNA was used in each digital PCR reaction. Mutant allele frequencies were calculated using QuantaSoft software (BioRad).

## Western blot

Proteins from cell lysates were separated electrophoretically using NuPAGE 4–12% Bis Tris Gels (Life technologies) in MOPS buffer (Life Technologies). Gels were then electroblotted onto Nitrocellulose Pre-Cut Blotting Membranes (Life Technologies), blocked with LI-COR blocking buffer (LI-COR), incubated with Rabbit anti-ERα antibody (SP-1; Thermo Scientific) and mouse Anti-α-Tubulin antibody (DM1A, Sigma-Aldrich). Membranes were washed with 0.1% tween–20 in PBS. Membranes were then incubated with IRDye 800 CW goat anti rabbit and IRDye 680 CW goat anti mouse secondary antibodies (LI-COR), washed, and scanned

using a LI-COR Odyssey infrared imaging system. ERα and α-tubulin levels were quantitated using a LI-COR Odyssey infrared imaging system. ER levels were normalized to α-tubulin. Percent ERα was defined as normalized ERα sample/normalized ERα untreated cells × 100.

## In vivo pharmacology

Animal studies were conducted in accordance with the Guide for the Care and Use of Laboratory Animals, National Academy Press (2006), conforming to California State legal and ethical practices and approved by the Institutional Animal Care and Use Committee (IACUC, Seragon and/or Genentech).

## Beeswax pellet manufacture

Beeswax pellets are prepared by mixing 50 mg 17α-Ethynylestradiol (Sigma) to 1.95 g melted beeswax (Sigma) @ 60°C for 10–15 min (until dissolved). Pellets are formed by applying 2–3 drops of wax/estradiol mixture onto a cold sterile weight boat using a glass Pasteur pipette. Procedure results in ~30 mg pellets containing ~1 mg estradiol, resulting in plasma estrogen levels of approximately 150pg/mL.

## Ovariectomy

With the animal in ventral recumbency, a 1 cm dorsal midline skin incision is made approximately 1 cm anterior to the base of the tail. A single incision of ~5 mm in length is made into the muscle wall on both the right and left sides and the ovaries and the oviducts are exteriorized. Each ovarian fat pad, ovary and part of the oviduct is excised with a cautery and the remaining tissue is returned to the peritoneal cavity. The skin incision is closed with wound clips.

## Xenograft models

For efficacy studies, drug treatments were initiated when tumors reached an average of 150–350 mm$^3$. Specific models are described below:

### WHIM20

The WHIM20 patient derived xenograft model (WHIM20) was established via direct implantation of metastatic tumor material into immune compromised mice at Washington University in St. Louis (*Li et al., 2013*). WHIM20 harbors an ESR1 Y537S mutation.

Tumors were established from fragments obtained from Washington University by subcutaneous implantation via 10G trocar into the right flank of 6–7 week old female NOD. CB17-Prkdc$^{scid}$/NcrCrl mice (Charles River Laboratories) and NOD.Cg-Prkdc$^{scid}$ Il2rg$^{tm1Wjl}$/SzJ mice (Jackson Laboratories). Tumor line was propagated by serial transplantation. Animals were ovariectomized prior to initiation of treatment.

For the pharmacodynamic (PD) experiment, tumor-bearing animals were treated with specified compounds for 4 days. Tumors were harvested 8 hr after the final dose and fixed in 10% formalin prior to paraffin-embedding, or flash-frozen in liquid nitrogen for subsequent protein and RNA extractions.

### MCF7 HA-ER.Y537S

Ovariectomized female Crl:NU-Foxn1$^{nu}$ mice (Charles River Laboratories) at 6–8 weeks of age implanted s.c. on the right flank with tumor fragments.

### TamR1

The tamoxifen-resistant derivative of the ER+ human tumor cell line, MCF7 (TamR1) was developed by Seragon Pharmaceuticals, Inc (*Lai et al., 2015*). Briefly, female Crl:NU-Foxn1$^{nu}$ mice bearing MCF7 tumors growing subcutaneously (harboring 0.72 mg/60 day release 17β-estradiol pellets; Innovative Research of America) were subjected to chronic daily tamoxifen

treatment (60 mg/kg/day) by oral gavage, resulting in an inhibition of tumor growth. Following emergence of tamoxifen resistant tumor growth the daily tamoxifen dose was increased to 120 mg/kg/day. The tumors which continued to grow under these conditions were serially passaged and evaluated for continued tamoxifen resistance. One tamoxifen-resistant tumor line developed by this method was designated TamR1. TamR1 expresses wild-type ERα and is dependent on exogenous estradiol or tamoxifen for growth in female Crl:NU-Foxn1$^{nu}$ mice.

TamR1 tumor fragments were implanted into the right lower flank of female Crl:NU-Foxn1$^{nu}$ mice (Charles River Laboratories). Animals were treated with tamoxifen until tumor fragments reached an average size of 200 mm$^3$, after which dosing with test article was initiated and continued for 28 days.

## HCI-003

The HCI-003 model was established at the University of Utah, Huntsman Cancer Center via direct implantation of both primary tumor material into immune compromised mice (*DeRose et al., 2011*). HCI-003 is wild-type at the ESR1 ligand binding domain. Exogenous estradiol is required for robust growth of this patient derived tumor line, and is supplied in the form of estradiol impregnated beeswax pellets. Tumor fragments obtained from the Huntsman Cancer Center were surgically implanted into the cleared mammary fat pad of 19 day old female NOD.CB17-Prkdc$^{scid}$/NcrCrl mice (Charles River Laboratories). The right mammary fat pad of recipient animals was exposed by a 1 cm incision on the ventral mid-line at the level of the fourth mammary. The blood vessels were cauterized and the fat pad blunt dissected from the abdominal wall and skin. Tumor fragments were excised from donor animals, chopped into fragments approximately 8 mm$^3$ (2 × 2 × 2 mm), implanted in the region of the cleared mammary fat pad, and the incision was closed with wound clips. Estradiol impregnated beeswax pellets were implanted subcutaneously into the interscapular space of recipient animals on the day of tumor implant. Recipient animals were ovariectomized 19 days after tumor fragment implant. One group of animals was designated as a control for estradiol independent tumor growth (Vehicle-E2) and the estradiol impregnated pellets were excised on the first day of dosing. All mice were housed and maintained according to the animal-use guidelines of the Institutional Animal Care and Use Committee (IACUC), conforming to California State legal and ethical practices.

## MCF7

Female Crl:NU-Foxn1$^{nu}$ mice (Charles River Laboratories) at 7 weeks of age were implanted s.c. with estradiol pellets (0.36 mg, 60 day release, Innovative Research) in the interscapular region. Three days later, 1 × 107 MCF7 cells/100 μL (in HBSS:Matrigel, 1: 1) were injected into the 2/3 mammary fat pad.

## ZR-75-1

Female Crl:NU-Foxn1$^{nu}$ mice (Charles River Laboratories) at 7 weeks of age were implanted s.c. with estradiol pellets (0.72 mg, 60 day release, Innovative research of America) in the interscapular region. Three days later, 5 × 10$^6$ ZR-75-1 cells/100 μL (in DMEM :Matrigel, 1 :1) were injected s.c. on the right flank.

## MDA-MB-231

Female Crl:NU-Foxn1$^{nu}$ mice (Charles River Laboratories) at 6 weeks of age were injected s.c. with 5 × 10$^6$ MDA-MB-231 cells/100 μL (in DMEM :Matrigel, 1 :1) on the right flank.

## FES-PET/CT Imaging Experiment - MCF7 Xenograft Tumors

### Cell lines

MCF7 cell line (ATCC) was maintained in IMEM/10% FBS (Gibco) and authenticated by short tandem repeat profiling using Sanger sequencing.

### Xenograft studies

Experiments with mice were approved by the Vanderbilt IACUC. Female 4- to 5-week old ovariectomized athymic Balb/c mice (Harlan Sprague Dawley) were implanted s.c. with a 14-day-release, 0.17 mg 17β-estradiol pellet (Innovative Research of America). The next day, $10^7$ MCF7 cells suspended in IMEM:Matrigel (BD Biosciences; 1:1 ratio) were injected s.c. in the right flank of each mouse. After 4 weeks, mice bearing tumors $\geq$250 mm$^3$ were randomized to treatment with vehicle (80 mM sodium citrate buffer, pH 3.0) or GDC-0810 (10 and 100 mg/kg/day, p.o.). Tumor diameters were measured using calipers twice per week and volume in mm$^3$ calculated with the formula: volume = width$^2$ × length/2. Tumors were harvested 1 hr after the last dose of GDC-0810 and either flash-frozen in liquid nitrogen or fixed in 10% formalin prior to paraffin-embedding. Five-μm paraffinized sections were used for immunohistochemistry (IHC) using ERα (Vector Labs, # VP-E613) and PR antibodies (Dako, # M3569). Sections were scored by a trained breast pathologist (M.V.E.) blinded to the type of treatment.

### FES-PET/CT imaging

On day 0 (pre-treatment baseline) and on day 7 (post-treatment), mice were anesthetized with 2% isofluorane and injected retrorbitally with ~11 MBq/0.1 mL of $^{18}$F-estradiol (specific activity ~9000 Ci/mmol). Sixty minutes later, mice were anesthetized with 2% isofluorane and imaged in an Inveon micro PET/CT (Siemens Preclinical Solutions, Knoxville, TN) for 20 min. All PET data sets were reconstructed using the MAP algorithm into 128 × 128 × 159 slices with a voxel size of 0.776 × 0.776 × 0.8 mm$^3$. The CT images were acquired at an x-ray beam intensity of 22 mAs and x-ray voltage of 80 kVp. The CT images were reconstructed into 768 × 768 × 512 voxels at a voxel size of 0.114 × 0.114 × 0.114 mm$^3$.

### Data analysis

Anatomical regions-of-interest (ROIs) were drawn around tumor and muscle (hind limb) in the CT images using the medical imaging analysis tool AMIDE and superimposed on the PET images (*Loening and Gambhir, 2003*). Radiotracer concentration within the tumor ROIs was divided by that of muscle to obtain a standard uptake value tumor-to-muscle ratio (SUVr) for each mouse. Post-treatment SUVr's in each vehicle- or GDC-0810-treated mouse were compared to the respective pre-treatment SUVr. Significant differences in post-treatment vs. baseline IHC histoscores and $^{18}$F-estradiol PET/CT imaging were determined by repeated ANOVA with Bonferroni post hoc-test. A p value of <0.05 was considered statistically significant.

### H-score

Tumors were harvested 1 hr after the last dose of GDC-0810 and flash-frozen in liquid nitrogen or fixed in 10% formalin prior to paraffin-embedding. Five-μm paraffinized sections were used for immunohistochemistry (IHC) using antibodies against ERα (Vector Labs, # VP-E613) and PR (Dako, # M3569). The histoscore for ERα and PR staining was assessed by an expert breast pathologist (M.V.E.) using a modified scoring method previously reported by Nenutil *et al*. Briefly, total cells and cells with positive staining were counted and the percent of positive nuclei over the total in each high power field was calculated. Fields were then assigned a relative staining intensity score of 1 for low, 2 for intermediate, or 3 for high staining. The product of the percent positive cells and staining intensity was then derived to create a histoscore of 0–300 for each high power field.

