## [Decision Letter]

Thank you for submitting your article "The selective estrogen receptor downregulator GDC-0810 is efficacious in diverse models of ER+ breast cancer" for consideration by *eLife*. Your article has been reviewed by four peer reviewers, and the evaluation has been overseen by a Reviewing Editor and Sean Morrison as the Senior Editor. The following individuals involved in the review of your submission have agreed to reveal their identity: John A Katzenellenbogen (Reviewer #1) and Myles Brown (Reviewer #2).

The reviewers have discussed the reviews with one another and the Reviewing Editor has drafted this decision to help you prepare a revised submission.

All of the reviewers agreed that your submitted manuscript presented some new results characterizing the mechanism of action of GDC-0810 and demonstrating its effectiveness in additional breast cancer models, including some work on two ER mutants that show clinical resistance to conventional endocrine therapies. The activity of GDC-0810, particularly on the mutants, although not extensive, provides evidence supporting its clinical translation.

Comments regarding the work on mechanisms focused on two points: First, there is an incomplete analysis of some of the results, which presumably could be readily improved. Second, a point of considerable discussion among reviewers involved the need to analyze the relationship between ER SERD and ER antagonist activity. This needs to be done in the context of the work by others in the field, but more importantly in the context of the in vitro vs. in vivo results presented in the manuscript itself: It is notable that GDC-0810, a SERD, and tamoxifen, an antagonist with no SERD activity, show comparable efficacy in suppressing xenograft growth. Hence, the statement, "Together, these in vitro results support the discovery strategy of optimizing ERα degradation in addition to ERα antagonism." appears to be contradicted by your results from the in vivo models. From what you present, one could conclude that beyond its superior PK and oral availability (which are clearly significant advances allowing higher dosing), GDC-0810 does not have mechanistic superiority over other SERDs and ER antagonists. This whole issue requires careful clarification and could benefit from additional work.

In the end, our recommendation is for "revision", and below is a summary of the major requested changes and additions.

Summary:

The authors report an interesting and quite thorough study characterizing the orally active SERD GDC-0810 as a potential drug treatment for metastatic breast cancer driven by mutant estrogen receptors. Their strategy is based on first identifying compounds that work as estrogen receptor degraders (SERDs) that were then studied further in a variety of ways to examine their efficacy, pharmacokinetic properties and oral bioavailability. There is a clear need for such drugs, since currently used agents that also degrade ER (Fulvestrant) have poor pharmacokinetic properties and must be given to breast cancer patients by intramuscular injection, a route that is painful and probably still gives an insufficient exposure of the tumors to the drug.

Major Revisions Requested:

1) Analyze and discuss more incisively the relationship between ER SERD and ER antagonist activity, including how in vitro results relate-or stand in contrast-to in vivo outcomes. GDC-0810 is likely to have activity in several contexts of hormone-resistance where ER dependence is retained. Work by other investigators using other SERDs and ER antagonists in these types of resistance models should be cited and discussed (Wardell, Ellis et al. 2015 in Clinical Cancer Research; Couillard et al. 1998 in Cancer Research). In particular, Discussion should address in some detail the Wardell paper as well as previous work by those authors (Wardell et al. 2013 Clinical Cancer Research, Wardell et al. 2011 Biochemical Pharmacology) regarding the question of whether degradation of ER by GDC-0810 is necessary for the drug's antitumor effects.

2) Provide a conclusion in which the factors of SERD and antagonist efficacy and potency, as well as PK, oral activity and accessible dosing are presented in a balanced way. In Figures 1 and 2, a distinction is made between GDC-0810 and Tamoxifen-like compounds on the basis of GDC-0810 inducing a different conformation and causing ER degradation, but then, in vivo, there is little difference between Tamoxifen and GDC-0810 in several xenografts (3A, 3C, 3D, 6B). It would be valuable if you could show a context (e.g., resistance due to Y537S) where the degradation matters in vivo. If not, you could simply conclude that at this point there is no correlation between ER degradation and in vivo activity, and that in vivo efficacy represents mainly potency of antagonism and improved PK.

3) Provide more discussion of mechanistic differences. In Figure 1E, 14 conformation selective peptides are studied with GDC-0810 and other SERM/SERD ER ligands. The findings nicely show that GDC-0810 induces a distinct conformation in wild type ERa, especially involving three of these peptides (GW5P2, 47P3, BN2). A more complete discussion of these intriguing results is needed:

A) Are these results dose-related and are they reproducible (unclear if there were biological or technical replicates) and are they confirmed by other assays?

B) Which conformation selective peptides (as representative of coregulators and other factors) were actually studied, and which showed changes or no changes in their interaction with wild type ER and GDC, vs. wild type ER and Fulvestrant, vs. wild type ER with E2? (This should be presented in a table in the manuscript or as a supplementary table.)

C) Can anything be learned from the three specific peptides (GW5P2, 47P3, BN2) whose interactions appear unique to GDC-0810 regarding the ultimate efficacy of GDC?

D) Peptide interaction studies with mutant ER Y537S or D538G vs. wt ER could be very informative.

E) Finally, it would be nice (though not absolutely required if strong responses can be made to the foregoing issues) to confirm these findings in cell lines with the appropriate coregulator proteins, and to test whether such unique interaction is necessary for GDC-0810 action as this would change the entire manuscript from being a largely descriptive study to one that includes a truly novel mechanistic observation.

Other Items Needing Addition or Correction:

4) The authors find that inhibition of the ER mutants requires significantly higher doses of GDC-0810 similar to what has been found for other ER ligands and shown by Fanning et al. to be the result of differences in affinity. What is the affinity of GDC-0810 for WT and mutant ERs?

5) Most of the studies of the ER mutants employ a single clone of MCF7 engineered using CRISPR to express the mutation. Is this clone representative of the entire population or multiple other clones? T47D cell engineered to express the mutation are not characterized in detail. Do these behave similarly to the MCF7 clones?

6) Abstract implies several points of 'novelty' that are of unclear significance. That GDC-0810 is the first orally bioavailable SERD in Phase 2 relies on a somewhat blurry definition of SERM vs. SERD vs. SERM/SERD. Does this represent a major distinction in expected antitumor activity from EM800 or bazedoxifene?

7) In several sections, attention to dose-dependence of biologic effects (e.g. Figure 2A/B) is not made, making it hard to interpret.

8) Figure 1C should include Fulvestrant as direct comparison.

9) Related to Figure 2A, it would be worth mentioning whether GDC-0810 is agonistic in other tissues such as uterus or bone.

10) Is the data shown in Figure 2C generated in the absence of presence of E2? The authors should include a dose response curve of GDC-0810 in the absence of E2. This should clearly be shown. The data shown in Figure 2C are not conclusive – it seems that GDC-0810 has some agonist activity at low doses (?). It can't be compared to Fulvestrant since that data looks inconsistent at low doses. This experiment should be repeated.

11) For in vivo treatment (e.g. Figure 3, and Figure 3—figure supplement 1B), it is assumed that the GDC-0810 treatment is given in the presence of E2 (it is described in more detail later in the manuscript). This should be more clearly indicated in both the figure and the legend.

12) It is not clear why the investigators choose to study interaction of the ESR1 mutant with PGCalpha (Figure 5A), which is a less well described co-activator in the context of ER and its mutants. Why didn't they test SRC1, SRC3 or other better-described co-activator interaction? (Or is the rationale for this experiment linked to data shown in Figure 1E? If that is the case, it should be stated).

13) It would be worth evaluating mutant and WT expression at the end of study in Figure 6A to evaluate transgene expression (mRNA) and degradation.

14) It is unclear when biological or technical replicates of experiments have been done. This should be more clearly stated (and possibly performed if not in some circumstances). Some figures lack statistical analysis (e.g. Figure 5).

15) IC50s for GDC-0810 and Fulvestrant for WT and mutant cells should be given.

16) What are PR levels in the ESR1 mutant tumor -/+ GDC-0810 treatment?

17) Wardell et al., 2013 reference should be completed.

18) The Methods are quite well written and thorough; however, it would be helpful if figure legends stated the number of days after cell injection or tumor fragment implantation that treatment with ligands was begun.

---

## [Author Response]

Comments regarding the work on mechanisms focused on two points: First, there is an incomplete analysis of some of the results, which presumably could be readily improved. Second, a point of considerable discussion among reviewers involved the need to analyze the relationship between ER SERD and ER antagonist activity. This needs to be done in the context of the work by others in the field, but more importantly in the context of the in vitro vs. in vivo results presented in the manuscript itself: It is notable that GDC-0810, a SERD, and tamoxifen, an antagonist with no SERD activity, show comparable efficacy in suppressing xenograft growth. Hence, the statement, "Together, these in vitro results support the discovery strategy of optimizing ERα degradation in addition to ERα antagonism." appears to be contradicted by your results from the in vivo models. From what you present, one could conclude that beyond its superior PK and oral availability (which are clearly significant advances allowing higher dosing), GDC-0810 does not have mechanistic superiority over other SERDs and ER antagonists. This whole issue requires careful clarification and could benefit from additional work.

We thank the reviewers for bringing to our attention what might be considered a disconnect between what we highlight as being a key feature in vitro (i.e. ER degradation) versus what appears to be sufficient for anti-tumor activity in vivo (i.e. ER antagonism). We have made significant investments in addressing the relationship between ER degradation and antagonist activity, including experimentally, by removing some data to de-emphasize some points, and by better placing our data and discussion in context of what has been previously reported. We demonstrate, using additional experimental data, that GDC-0810’s ability to suppress ER signaling can be uncoupled from its ability to deplete ER, and acknowledge here, and in the Discussion that its potent ER antagonism coupled with good PK and bioavailability are likely all major contributing factors to robust in vivo efficacy, potentially independent of degradation. We also note though, in the context of newly included uterus data, that ER degradation may be associated with preventing ER agonism, which may be more relevant for long-term/acquired resistance in vivo. We believe the manuscript has greatly benefited from clarification surrounding this important issue (specific details below).

*Summary:*

*The authors report an interesting and quite thorough study characterizing the orally active SERD GDC-0810 as a potential drug treatment for metastatic breast cancer driven by mutant estrogen receptors. Their strategy is based on first identifying compounds that work as estrogen receptor degraders (SERDs) that were then studied further in a variety of ways to examine their efficacy, pharmacokinetic properties and oral bioavailability. There is a clear need for such drugs, since currently used agents that also degrade ER (Fulvestrant) have poor pharmacokinetic properties and must be given to breast cancer patients by intramuscular injection, a route that is painful and probably still gives an insufficient exposure of the tumors to the drug.*

*Major Revisions Requested:*

1) Analyze and discuss more incisively the relationship between ER SERD and ER antagonist activity, including how in vitro results relate-or stand in contrast-to in vivo outcomes. GDC-0810 is likely to have activity in several contexts of hormone-resistance where ER dependence is retained. Work by other investigators using other SERDs and ER antagonists in these types of resistance models should be cited and discussed (Wardell, Ellis et al. 2015 in Clinical Cancer Research; Couillard et al. 1998 in Cancer Research). In particular, Discussion should address in some detail the Wardell paper as well as previous work by those authors (Wardell et al. 2013 Clinical Cancer Research, Wardell et al. 2011 Biochemical Pharmacology) regarding the question of whether degradation of ER by GDC-0810 is necessary for the drug's antitumor effects.

We have used the “ER over-expression strategy” described by Wardell et al. (2013) Clinical Cancer Research and Wardell et al. (2011) Biochemical Pharmacology in order to test whether degradation of ER by GDC-0810 is necessary for the drug’s antagonistic properties in vitro. These data are provided as new panels in Figure 2C, D, E, and are described and discussed in the last paragraph of subsection “GDC-0810 induces a distinct ERα conformation versus tamoxifen and other ER therapeutics, and does not exhibit tamoxifen-like ER agonism in MCF7 cells”, and in the second and last paragraphs of the Discussion. We find, similar to what had previously been reported for fulvestrant and bazedoxifene, that the depletion of ER is neither required for the ability of GDC-0810 to suppress the expression of PR, as a read-out for ER pathway activity, nor for its ability to suppress the proliferation of MCF7 cells. This is entirely consistent with the demonstration that GDC-0810 acts directly as an ER antagonist, by outcompeting E2 and changing the ER conformation such that the PGC1α co-activator, for example, is displaced, in cell free assays where degradation does not occur. We note here that while over-expression of ER had previously been seen to block ER degradation mediated by fulvestrant, in our experiment fulvestrant retained its degradation activity. We speculate that the overexpression of ER in our doxycycline-inducible cells may not be as high as what was reported previously – though these levels are sufficient to prevent GDC-0810’s ability to deplete ER, consistent with distinct mechanisms of degradation driven by fulvestrant versus GDC-0810.

2) Provide a conclusion in which the factors of SERD and antagonist efficacy and potency, as well as PK, oral activity and accessible dosing are presented in a balanced way. In Figures 1 and 2Figure 1, a distinction is made between GDC-0810 and Tamoxifen-like compounds on the basis of GDC-0810 inducing a different conformation and causing ER degradation, but then, in vivo, there is little difference between Tamoxifen and GDC-0810 in several xenografts (3A, 3C, 3D, 6B). It would be valuable if you could show a context (e.g., resistance due to Y537S) where the degradation matters in vivo. If not, you could simply conclude that at this point there is no correlation between ER degradation and in vivo activity, and that in vivo efficacy represents mainly potency of antagonism and improved PK.

We acknowledge, and agree with the reviewers’ point that despite clear mechanistic differences in vitro, GDC-0810 and tamoxifen both exhibit anti-tumor activity in most of our pre-clinical models in vivo, which supports the idea that in these cases, in vivo efficacy represents mainly potency of antagonism and good drug-like properties. We propose that the advantage of the SERD approach will be most clearly highlighted in cases that represent re-activation of ER signaling in the presence of therapeutics, illustrated by the TamR1 model for example. It has been argued that clinically, ER-dependent resistance mechanisms to both AI’s and SERMs account for a high fraction of resistant and relapsing disease. Unfortunately, our pre-clinical models do not represent this scenario, and are far from representative of the enormous heterogeneity of ER-dependent resistance mechanisms likely to be found in the clinic. We argue that truly understanding whether the SERD approach will drive meaningful benefit over what can be achieved with antagonist/tamoxifen-like approaches will be unveiled only through rigorous clinical testing in that heterogeneous population. Recent clinical trial results indicate that fulvestrant is superior to aromatase inhibitor (FALCON) and prior trials indicate that aromatase inhibitors and equal to or superior to tamoxifen. Since these points provide important context for this manuscript we have now tried to be considerably more direct, clear and balanced in this respect, in particular in our revised Discussion (last paragraph).

*3) Provide more discussion of mechanistic differences. In Figure 1E, 14 conformation selective peptides are studied with GDC-0810 and other SERM/SERD ER ligands. The findings nicely show that GDC-0810 induces a distinct conformation in wild type ERa, especially involving three of these peptides (GW5P2, 47P3, BN2). A more complete discussion of these intriguing results is needed:*

We have provided additional data, details, as well as a more complete discussion of our findings related to the conformational profiling assay, as requested, see subsection “GDC-0810 induces a distinct ERα conformation versus tamoxifen and other ER therapeutics, and does not exhibit tamoxifen-like ER agonism in MCF7 cells” and the third paragraph of the Discussion. Specific points raised by the reviewers are addressed below:

*A) Are these results dose-related and are they reproducible (unclear if there were biological or technical replicates) and are they confirmed by other assays?*

Since the goal of the experiment was to evaluate potential differences in ER conformation induced by various ligands, and each of the ligands has a different binding affinity for ER, we used a single, saturating dose of ligand (1μM), to neutralize those differences in binding; these findings are therefore unlikely to be dose-related. We have added this information to the figure legend for Figure 2, to the description in the Results in the subsection “GDC-0810 induces a distinct ERα conformation versus tamoxifen and other ER therapeutics, and does not exhibit tamoxifen-like ER agonism in MCF7 cells”, and to the supplementary methods subsection “ERα Conformational Profiling Assay”. This approach is in keeping with how similar conformational profiling assays have been used both historically (Iannone et al., 2004; Norris et al., 1999) and more recently (Wardell et al., 2013). Data provided for these experiments are from biological triplicates and is now specified in the manuscript.

*B) Which conformation selective peptides (as representative of coregulators and other factors) were actually studied, and which showed changes or no changes in their interaction with wild type ER and GDC, vs. wild type ER and Fulvestrant, vs. wild type ER with E2? (This should be presented in a table in the manuscript or as a supplementary table.)*

The data provided shows all peptides and drug-induced ER interactions that we studied using this approach, all in the context of wild-type ER. The only other co-regulator interaction that was studied in the context of GDC-0810 is PGC1α, and that data is provided in Figure 6B,though this is a different assay format and question that is being addressed. The majority of these peptides was derived from phage display approaches, and are not representative of naturally-occurring co-regulators or other factors (see also below).

*C) Can anything be learned from the three specific peptides (GW5P2, 47P3, BN2) whose interactions appear unique to GDC-0810 regarding the ultimate efficacy of GDC?*

We completely agree that the very robust interaction of the GDC-0810:ER complex with peptides GW5P2, 47P3, BN2 is very intriguing. We have provided additional data in Figure 2A demonstrating that these same peptides robustly interact with the GW7604:ER complex. Peptides GW5P2 and 47P3 were experimentally derived though phage display, as peptides that would specifically interact with this complex. It was reported that the presence of a carboxylic or acrylic acid in the head group of a large panel of ER ligands correlated with the recruitment of these peptides to ER. Recruitment of these same peptides to the GDC-0810:ER complex therefore likely reflects the presence of an acrylic acid in GDC-0810. We now provide these details, and further elaborate in our revised Discussion (third paragraph).

*D) Peptide interaction studies with mutant ER Y537S or D538G vs. wt ER could be very informative.*

We agree that this would be an interesting experiment, though would be more suited to a manuscript in which the main focus is the biology of mutant ER, rather than this manuscript in which we are aiming to focus on characterization of GDC-0810. We have, where possible, tried to contribute to this important discussion by generating and characterizing 2 distinct ER knock in mutant models. Indeed, we believe that such knock-in cell lines are the first to be described for the ESR1 mutations and as such are a meaningful contribution to the community.

E) Finally, it would be nice (though not absolutely required if strong responses can be made to the foregoing issues) to confirm these findings in cell lines with the appropriate coregulator proteins, and to test whether such unique interaction is necessary for GDC-0810 action as this would change the entire manuscript from being a largely descriptive study to one that includes a truly novel mechanistic observation.

We would love to be able to probe the functional relevance of these peptide interactions in order to gain a deeper mechanistic understanding of GDC-0810, as the reviewers suggest. Unfortunately, these peptides are not derived from naturally occurring proteins (see also above), which we could experimentally manipulate. We plan to pursue proteomics approaches to try and identify endogenous members of the ER:GDC-0810 complex, but feel that such data would be beyond the scope of this manuscript.

*Other Items Needing Addition or Correction:*

4) The authors find that inhibition of the ER mutants requires significantly higher doses of GDC-0810 similar to what has been found for other ER ligands and shown by Fanning et al. to be the result of differences in affinity. What is the affinity of GDC-0810 for WT and mutant ERs?

We have provided additional ER.WT and mutant competitive binding data in Figure 6A; there is a small shift in affinity, as expected according to Fanning et al.

*5) Most of the studies of the ER mutants employ a single clone of MCF7 engineered using CRISPR to express the mutation. Is this clone representative of the entire population or multiple other clones?*

We have provided additional data in Figure 6—figure supplement 1; our observations are very similar across multiple ER.Y537S clones.

T47D cell engineered to express the mutation are not characterized in detail. Do these behave similarly to the MCF7 clones?

We have provided additional data as a new Figure 7, and described in the fourth paragraph of the subsection “GDC-0810 antagonizes ERα ligand binding domain mutants in vitro and in vivo”, showing characterization of key features of the T47D ER.D538G cells relative to their ER.WT parental cells, and also provide an additional cell viability experiment. We see similar trends as in the MCF7 ER.Y537S clones in terms of estrogen-independent pathway activity and a potency shift for GDC-0810 in proliferation assays.

6) Abstract implies several points of 'novelty' that are of unclear significance. That GDC-0810 is the first orally bioavailable SERD in Phase 2 relies on a somewhat blurry definition of SERM vs. SERD vs. SERM/SERD. Does this represent a major distinction in expected antitumor activity from EM800 or bazedoxifene?

We have removed the word “first” from the Abstract to avoid this potential issue.

7) In several sections, attention to dose-dependence of biologic effects (e.g. Figure 2A/B) is not made, making it hard to interpret.

We have made great efforts to perform dose concentration curves where appropriate, and where logistically reasonable. The rationale for selecting a single fixed dose for Figure 2A is described above. For Figure 2B, we likewise selected a single high dose, across a number of genes, with a collection of positive controls in an attempt to screen for agonistic activity of GDC-0810 in MCF7 cells. It seems that at least 1 reviewer is particularly interested in whether we see any agonistic activity for GDC-0810 at low concentrations (see also point 10 below) perhaps triggered by the observation that some ER ligands can exert a biphasic response on ER signaling. We have therefore included additional data here, showing results from a gene expression analysis (Taqman) as a dose response curve. Unlike what we find for 4OH-tamoxifen, we do not observe evidence of agonism triggered by GDC-0810 in MCF7 cells, regardless of concentration or presence versus absence of E2 (see Author response image 1). Though such lack of agonism may be tissue context dependent, since we do see weak agonism in Ishikawa endometrial cells.

**Author response image 1. respfig1:** MCF7 cells were treated with increasing concentrations of either GDC-0810 (left) or 4OH-tamoxifen (right), for 6 hours, prior to analysis of PGR and GREB1 as ER target genes, by Taqman assays. Gene expression was normalized to DMSO control (set to 1) either in the presence or absence of 1nM E2.

8) Figure 1C should include Fulvestrant as direct comparison.

We have included this as Figure 1D.

9) Related to Figure 2A, it would be worth mentioning whether GDC-0810 is agonistic in other tissues such as uterus or bone.

We have provided additional data, as Figure 3A–F, detailing the activity of GDC-0810 in Ishikawa (endometrial) cells in vitro, and in the rat uterus in vivo. We find that GDC-0810 exhibits mild estrogenic activity in this context. We note that in this context, GDC-0810 does not exhibit robust ER depletion, though fulvestrant, which acts as an inverse agonist does degrade ER. These observations are described in the subsection “GDC-0810 displays mild estrogenic activity in uterine models in vitro and in vivo” and in the second paragraph of the Discussion.

10) Is the data shown in Figure 2C generated in the absence of presence of E2? The authors should include a dose response curve of GDC-0810 in the absence of E2. This should clearly be shown. The data shown in Figure 2C are not conclusive – it seems that GDC-0810 has some agonist activity at low doses (?). It can't be compared to Fulvestrant since that data looks inconsistent at low doses. This experiment should be repeated.

We have provided a dose response curve for GDC-0810 (as well as fulvestrant and 4OH-tamoxifen) in both the absence and the presence of E2, as requested, in Figure 1E, F. We see no clear evidence of agonism using cell proliferation as a read-out, including for 4OH-tamoxifen, for which we see some agonism at the ER pathway level, though only for a sub-set of target genes.

11) For in vivo treatment (e.g. Figure 3, and Figure 3—figure supplement 1B), it is assumed that the GDC-0810 treatment is given in the presence of E2 (it is described in more detail later in the manuscript). This should be more clearly indicated in both the figure and the legend.

This information had been provided in the figure legend as, for example “in the presence of 60-day release 0.36 mg 17β-estradiol pellets”, and “HCI-003 patient derived xenograft tumors were implanted in mice containing a 1 mg 17β-estradiol beeswax pellet”. The presence of E2 is indicated in the figure itself as “Vehicle (+E2)” versus “Vehicle (-E2)”.

12) It is not clear why the investigators choose to study interaction of the ESR1 mutant with PGCalpha (Figure 5A), which is a less well described co-activator in the context of ER and its mutants. Why didn't they test SRC1, SRC3 or other better-described co-activator interaction? (Or is the rationale for this experiment linked to data shown in Figure 1E? If that is the case, it should be stated).

We originally profiled a collection of ER ligands (not including GDC-0810), against a number of co-activator peptides in the context of WT and mutant ER. Though trends were similar across these peptides, the absolute signal and dynamic range varied. We thus selected PGC1α for further studies since it reflected the general trends, had robust signal and dynamic range, and was also commercially available in a format amenable to our studies in a large number of compounds.

13) It would be worth evaluating mutant and WT expression at the end of study in Figure 6A to evaluate transgene expression (mRNA) and degradation.

Since this model (previously Figure 6A, now Figure 8A) is an artificially engineered model, over-expressing mutant ER under the non-endogenous EF1 promoter, we chose to perform pharmacodynamic/response studies in the WHIM20 PDX model instead. These new data are presented in Figure 8B–D.

14) It is unclear when biological or technical replicates of experiments have been done. This should be more clearly stated (and possibly performed if not in some circumstances). Some figures lack statistical analysis (e.g. Figure 5).

We have tried to be explicit about replicates and have added the requested statistical analyses.

15) IC50s for GDC-0810 and Fulvestrant for WT and mutant cells should be given.

We have added IC50 values to all of our dose response curves.

16) What are PR levels in the ESR1 mutant tumor -/+ GDC-0810 treatment?

We have included western blot, RPPA as wells as IHC data, evaluating levels of PR as Figure 8C, D.

17) Wardell et al., 2013 reference should be completed.

References have been updated.

*18) The Methods are quite well written and thorough; however, it would be helpful if figure legends stated the number of days after cell injection or tumor fragment implantation that treatment with ligands was begun.*

Our standard procedure for efficacy studies is to group tumor bearing animals based on tumor size (an average of 150–300mm^3^), prior to initiating dosing, rather starting dosing based on time since implantation, since the time for tumor initiation can vary widely from animal to animal. This information is included for specific models in the Methods, last paragraph,Figure 8 legend, subsections “TamR1” and “Xenograft Studies” and Figure 4—figure supplement 2 legend.